# Impacts of coagulation on the appearance time method for new particle growth rate evaluation and their corrections

Runlong Cai[1,2,*], Chenxi Li[3], Xu-Cheng He[2], Chenjuan Deng[4], Yiqun Lu[5], Rujing Yin[4], Chao Yan[1], Lin Wang[5], Jingkun Jiang[4], Markku Kulmala[1,2], Juha Kangasluoma[1,2]

[1]Aerosol and Haze Laboratory, Beijing Advanced Innovation Center for Soft Matter Science and Engineering, Beijing University of Chemical Technology, 100029 Beijing, China
[2]Institute for Atmospheric and Earth System Research / Physics, Faculty of Science, University of Helsinki, 00014 Helsinki, Finland
[3]School of Environmental Science and Engineering, Shanghai Jiao Tong University, 200240, China
[4]State Key Joint Laboratory of Environment Simulation and Pollution Control, School of Environment, Tsinghua University, 100084 Beijing, China
[5]Shanghai Key Laboratory of Atmospheric Particle Pollution and Prevention (LAP[3]), Department of Environmental Science and Engineering, Fudan University, Shanghai 200433, China

*Correspondence to*: Runlong Cai (runlong.cai@helsinki.fi)

*Keywords*: particle growth rate; coagulation; appearance time; new particle formation; sub-3 nm aerosol, clustering

**Abstract.** The growth rate of atmospheric new particles is a key parameter that determines their survival probability to become cloud condensation nuclei and hence their impact on the climate. There have been several methods to estimate the new particle growth rate. However, due to the impact of coagulation and measurement uncertainties, it is still challenging to estimate the initial growth rate of new particles, especially in polluted environments with high background aerosol concentrations. In this study, we explore the influences of coagulation on the appearance time method to estimate the growth rate of sub-3 nm particles. The principle of the appearance time method and the impacts of coagulation on the retrieved growth rate are clarified via derivations. New formulae in both discrete and continuous spaces are proposed to correct the impacts of coagulation. Aerosol dynamic models are used to test the new formulae. New particle formation in urban Beijing is used to illustrate the importance to consider the impacts of coagulation on sub-3 nm particle growth rate and its calculation. We show that the conventional appearance time method needs to be corrected when the impacts of coagulation sink, coagulation source, and particle coagulation growth are non-negligible compared to the condensation growth. Under the simulation conditions with a constant concentration of non-volatile vapors, the corrected growth rate agrees with the theoretical growth rates. However, the uncorrected parameters, e.g., vapor evaporation and the variation of vapor concentration, may impact the growth rate obtained with the appearance time method. Under the simulation conditions with a varying vapor concentration, the average bias of the corrected 1.5-3 nm particle growth rate range from 6-44%, and the maximum bias of size-dependent growth rate is 150%. During the test new particle formation event in urban Beijing, the corrected condensation growth rate of sub-3 nm particles was in accordance with the growth rate contributed by sulfuric acid condensation, whereas the conventional appearance time method overestimated the condensation growth rate of 1.5 nm particles by 80%.

## 1 Introduction

New particle formation (NPF) is frequently observed in various atmospheric environments (Kulmala et al., 2004; Kerminen et al., 2018; Nieminen et al., 2018; Lee et al., 2019). It contributes significantly to the number concentrations of aerosol and cloud condensation nuclei (CCN) and hence impacts the global climate (Kuang et al., 2009; Kerminen et al., 2012). New particle growth rate is one of the key parameters to characterize NPF events. On the one hand, the newly formed particles (~1 nm) have to survive from coagulation scavenging before they grow to the CCN size (~100 nm). Given the same background aerosol concentration, i.e., the same coagulation loss rate, it is the growth rate that determines the survival probability of new particles (Weber et al., 1997; Lehtinen et al., 2007). Therefore, measuring new particle growth rate accurately contributes to understanding the impact of NPF on the climate. On the other hand, particle growth rate is a key to investigate the growth mechanisms. Theoretical particle growth rates contributed by condensing vapors are usually compared to measured growth rates to reveal the possible particle growth mechanisms (Ehn et al., 2014; Yao et al., 2018; Mohr et al., 2019). A non-biased and accurate determination of measured growth rates is an important fundament of these comparisons.

Although new particle growth rates are frequently reported in various environments around the world, it remains difficult to retrieve accurate particle growth rates from an ambient dataset. Due to the varying atmospheric conditions, significant Kelvin effect, and size-dependent particle compositions, particle growth rate is a function of both time and particle size. The measured evolution of aerosol size distribution does not directly indicate the size-and-time-resolved growth rate of single particles because one cannot directly track single particles from the size distributions. There are several methods to obtain the size-and-time resolved growth rate by solving aerosol general dynamic equations (GDE, Kuang et al., 2012; Pichelstorfer et al., 2018). However, few applications of these GDE methods have been reported for particle growth analysis in the real atmosphere (e.g., Kuang et al., 2012). The most likely reason is that these GDE methods are sensitive to measurement uncertainties caused by atmospheric instability and instruments, which needs to be solved in future studies.

Apart from solving the GDEs, the widely used methods to estimate particle growth rate are based on finding the representing particle diameter or time. The representing diameter method usually uses the peak diameter of the size distribution of new particles and estimates its increase rate from its temporal evolution. The rate of increase of peak diameter is then taken as particle growth rate (Kulmala et al., 2012) after correcting (or sometimes neglecting) the influence of coagulation on the peak shifting (Stolzenburg et al., 2005). During the correction, coagulation is often classified into innermodal coagulation (self-coagulation) and intermodal coagulation (Anttila et al., 2010; Kerminen et al., 2018). The peak diameter is usually obtained by fitting a lognormal function to the measured aerosol size distribution of new particles. With a distinct peak diameter in the growing particle population, this method is theoretically feasible to estimate new particle formation rates. However, the mode fitting is usually tricky, especially when there is no well-defined mode in the growing distribution, either due to the aerosol distribution itself or the measurement uncertainties.

The representing time method estimates the corresponding time for a series of diameters according to a certain criterion and then calculates the growth rate according to the relationship between the diameters and their corresponding time (Dada et

al., 2020). The corresponding time is determined as the time to reach either the maximum concentration (maximum concentration method, Lehtinen and Kulmala, 2003) or a certain proportion of the maximum concentration (appearance time method, Lehtipalo et al., 2014) of a given particle size bin. Previous studies have tested the appearance time under various modeling conditions. Their results indicate that some appearance time methods can reproduce the theoretical growth rate within

acceptable uncertainties under certain test conditions (Lehtipalo et al., 2014) but not under other test conditions (Olenius et al., 2014; Kontkanen et al., 2016; Li and McMurry, 2018). As shown in the Theory section below, the discrepancy is because the slope of particle size against their appearance time usually convolves other information (e.g., coagulation) in addition to particle growth.

Determining the growth rate of sub-3 nm particles is more challenging than that of larger particles. Firstly, there are

considerable uncertainties in the measured sub-3 nm aerosol size distributions (Kangasluoma et al., 2020) compared to larger-sized particles (e.g., > 10 nm, Wiedensohler et al., 2012). These uncertainties pose a great challenge to the methods based on solving aerosol general dynamic equations. Secondly, during a typical atmospheric NPF event, the sub-3 nm particle size distribution function usually decreases monotonically with the increasing diameter (Jiang et al., 2011b). As a result, the representing diameter method is usually difficult to cover the sub-3 nm size range. In contrast, despite lacking a clear

mathematical understanding of the information convolved in the slope of appearance time against particle diameter, the appearance time method is usually favored for sub-3 nm particles and clusters because of the existence of the concentration peak of new particles during an atmospheric NPF event. In addition, the appearance time method is not significantly affected by the systematic instrumental uncertainties because the appearance time of each size bin is only determined by the relative signal rather than the absolute particle concentration.

Coagulation impacts both particle growth and the growth rate calculation, especially for polluted environments and some chamber studies with high aerosol concentrations. The impact of coagulation on aerosol dynamics has been known since decades ago (e.g., McMurry, 1983). Recent studies discussed the importance of considering coagulation when estimating new particle growth rate (Cai and Jiang, 2017), the influence of transport on measured size distributions (Cai et al., 2018), and primary particle emissions (Kontkanen et al., 2020) under a high aerosol concentration. Similarly, neglecting particle

coagulation may cause a bias in the retrieved particle growth rate. Therefore, the coagulation growth has to be considered before investigating the contributions of various condensing vapors to particle growth.

In this study, the feasibility and limitations of the appearance time method are investigated based on theoretical derivations. The impact of coagulation on the retrieved growth rate using the appearance time method is explored and then corrected. Aerosol dynamic models are used to test the conventional and corrected methods. After that, the corrected appearance time

method is applied in a typical NPF event in urban Beijing to show the impact of coagulation on growth rate evaluation in the real atmosphere. In addition, the potential uncertainties of the corrected appearance time method cause by vapor evaporation and the variation of vapor concentration and are discussed.

## 2 Theory

### 2.1 Particle growth rate

Before deriving the formulae for the appearance time method, the definitions of particle growth and coagulation loss have to be clarified to avoid potential misunderstanding. Although widely used in NPF analyses, the exact meanings of these two concepts vary with their applied conditions.

Particle growth rate, by definition, is the rate of increase in particle diameter as a function of time for a given particle. Assuming that there is a sufficient number of particles of the same size and compositions, it is reasonable to neglect the influence of the stochastic effect due to a low particle number and use the expectation of the single-particle growth rate to characterize the growth of the aerosol population with the same size. When there is only one non-volatile condensing vapor,

10 the formula for the expectation of the single-particle condensation growth rate (referred to as the condensation growth rate below for simplicity) is shown in Eq. 1:

$$\text{GR}_{\text{cond}} = \frac{\Delta d_{\text{p}}}{\Delta t} = \beta_{1,\text{p}} N_1 \cdot \left[ \sqrt[3]{\left( d_{\text{p}}{}^3 + d_1{}^3 \right)} - d_{\text{p}} \right] \tag{1}$$

where $\text{GR}_{\text{cond}}$ is the condensation growth rate ($\text{nm} \cdot \text{s}^{-1}$) that neglects evaporation, $d_{\text{p}}$ is particle diameter (nm), $t$ is time (s), $d_1$ is the diameter of the condensing vapor (nm), $\beta_{1,\text{p}}$ is the coagulation coefficient between $d_1$ and $d_{\text{p}}$ ($\text{cm}^3 \cdot \text{s}^{-1}$), and $N_1$ is the vapor concentration ($\text{cm}^{-3}$). Particle evaporation is assumed to be negligible and the particle shape is assumed to be spherical both

15 before and after the growth. Note that Eq. 1 is expressed in the discrete form, i.e., it does not assume a continuum particle size ($d_1 \rightarrow 0$). When multiple vapors contribute to particle growth simultaneously, the total condensation growth rate is the sum of the condensation growth rates contributed by every single vapor. When considering particle evaporation, i.e., monomer dissociation, particle growth due to the net effect of vapor association and dissociation will be explicitly referred to as net condensation growth.

In addition to the condensation of vapors, coagulation also contributes to particle growth. For a given particle with the size of $d_{\text{p}}$, the coagulation with a particle much smaller than $d_{\text{p}}$ is usually considered as a contribution to its growth. In contrast, the coagulation with a particle much larger than $d_{\text{p}}$ is usually considered as the coagulation loss of particle $d_{\text{p}}$. We follow this convention to distinguish coagulation growth and loss, i.e., particle coagulation with another particle no larger than itself is taken as coagulation growth and otherwise, it is taken as coagulation loss. Hence, the formula for the expectation of single-

25 particle coagulation growth rate (referred to as the coagulation growth rate for simplicity) in the discrete form is:

$$\text{GR}_{\text{coag}} = \sum_{d_{\text{i}}=d_{\text{min}}}^{d_{\text{i}}=d_{\text{p}}} \left\{ \beta_{\text{p,i}} N_{\text{i}} \cdot \left[ \sqrt[3]{\left( d_{\text{p}}{}^3 + d_{\text{i}}{}^3 \right)} - d_{\text{p}} \right] \right\} \tag{2}$$

where $\text{GR}_{\text{coag}}$ is the coagulation growth rate ($\text{nm} \cdot \text{s}^{-1}$), $d_{\text{min}}$ is the minimum particle size (nm), $\beta_{\text{p,i}}$ is the coagulation coefficient ($\text{cm}^3 \cdot \text{s}^{-1}$) between $d_{\text{p}}$ and $d_{\text{i}}$, and $N_{\text{i}}$ is the concentration ($\text{cm}^{-3}$) of particles with the size $d_{\text{i}}$. Since both condensation and coagulation contribute to particle growth, the total single-particle growth rate is equal to the sum of $\text{GR}_{\text{cond}}$ and $\text{GR}_{\text{coag}}$.

When retrieving particle growth rate from the measured aerosol size distributions, the retrieved value is named the apparent growth rate. "Apparent" emphasizes that the method does not necessarily guarantee that the retrieved growth rate is equal to the condensation or total growth rate of a single particle or the investigated aerosol population. When using the representing diameter method, the retrieved apparent growth rate is the increasing rate of the peak diameter and it does not directly characterize the growth of any particle(s). For instance, the coagulation loss rate is a function of particle diameter; as a result, the peak diameter shifts towards larger sizes with time because smaller particles are scavenged faster by coagulation than larger particles. Similarly, other size-dependent processes such as condensation and coagulation growth also cause the shift of peak diameter. As a result, the apparent growth rate sometimes needs to be corrected before taken as the total growth rate or the condensation growth rate (Stolzenburg et al., 2005). When using the representing time method, although the retrieved apparent growth rate is close to the condensation growth rate under some modeling conditions (Lehtipalo et al., 2014), their deviation can be significant under other conditions, on which we elaborate in section 4.

## 2.2 Coagulation sink and source

For a given particle, its coagulation with another particle can be classified into coagulation growth and coagulation loss as aforementioned. This classification is based on the Lagrangian specification that tracks the growth of a single particle. In contrast, according to the Eulerian specification that focuses on given particle diameters, each coagulation causes a sink of two particles and a source of one new particle with a larger diameter regardless of the particle sizes. Herein, we define the coagulation sink and source as the loss and production rate for particle size bins in the Eulerian specification. According to these definitions, the coagulation of a large particle with another smaller particle is counted as the coagulation sink (in the Eulerian specification) but not as the coagulation loss (in the Lagrangian specification) of the large particle. Following previous studies, we use CoagS ($s^{-1}$) to represent the sink coefficient (Kulmala et al., 2001) and CoagSrc ($cm^{-3} \cdot s^{-1}$) to represent the production rate due to coagulation (Kuang et al., 2012). Their formulae in the discrete form are given below:

$$\text{CoagS} = \sum_{d_i=d_{min}}^{d_i=d_{max}} \beta_{p,i} N_i \tag{3}$$

$$\text{CoagSrc} = \sum_{d_i=d_{min}}^{d_i^3=d_p^3-d_{min}^3} 0.5 \beta_{i,j} N_i N_j \tag{4}$$

where $d_{max}$ is the maximum particle diameter (nm); $d_j$ is defined by $d_j^3 = d_p^3 - d_i^3$; $N_i$ and $N_j$ are the concentrations of $d_i$ and $d_j$, respectively; and other variables have been introduced above. Note that CoagS and CoagSrc are defined differently and their units are also different. Table 1 summarizes the differences between the definitions in the Lagrangian and Eulerian specifications.

**Table 1**

## 2.3 Formulae for the new appearance time method

The conventional appearance time method has been discussed in detail in Lehtipalo et al. (2014) and its derivation under ideal conditions has been reported in He et al. (2020). Here we briefly describe the procedure to retrieve particle growth rate from the temporal evolution of a measured aerosol size distribution using the appearance time method. For each aerosol size bin, its corresponding appearance time is determined as the moment that the measured aerosol concentration in this size bin reached 50% or any other given proportion of its maximum concentration during the event. The maximum and half-maximum concentration of this size bin can either be taken from the smoothed temporal evolution of the measured concentration or determined by fitting a sigmoid function to the measured data. The growth rate is then estimated as the slope of the diameter of aerosol size bins versus their corresponding appearance time, i.e.,

$$GR_{conv} = \frac{\Delta d_p}{\Delta t} \tag{5}$$

where $GR_{conv}$ is the total growth rate (nm·s$^{-1}$) retrieved by the conventional appearance time method; $\Delta d_p$ (nm) is the size difference between two adjacent measured size bins, and $\Delta t$ (s) is the time difference of the appearance time of these two size bins. Note that since the appearance time is a function of $d_p$ and $GR_{conv}$ is estimated for different appearance times, the apparent relationship between $GR_{conv}$ and $d_p$ does not necessarily indicate the size dependency of $GR_{conv}$ at any given moment.

The correction formulae for influences of coagulation on the appearance time method in the discrete space is:

$$GR_{corr,tot} = GR_{conv} - \left( CoagS + \frac{CoagSrc}{2N_p} \right) \cdot \left[ \sqrt[3]{(d_p^{\ 3} + d_1^{\ 3})} - d_p \right] \tag{6}$$

$$GR_{corr,cond} = GR_{corr,tot} - GR_{coag} \tag{7}$$

where $GR_{corr,tot}$ is the total growth rate (nm·s$^{-1}$) after correcting the impact of both coagulation sink and source; $GR_{corr,cond}$ is the condensation growth rate (nm·s$^{-1}$) after correction; $GR_{coag}$ is the coagulation growth rate (nm·s$^{-1}$); CoagS (s$^{-1}$) and CoagSrc (cm$^{-3}$·s$^{-1}$) are the coagulation sink and coagulation source term for $d_p$, respectively; $N_p$ is the number concentration of particles with the size $d_p$ at its appearance time. Note that coagulation growth is corrected in Eq. 7 but not Eq. 6.

When measuring aerosol size distribution using size spectrometers, the measured distributions are usually reported in a certain number of sectional bins. Therefore, in addition to the formula in the discrete form (Eq. 6), the correction formula for the appearance time in the sectional form is given below:

$$GR_{corr,tot} = GR_{conv} - \left( CoagS + \frac{CoagSrc}{2N_{[dp,l\ dp,u]}} \right) \cdot \left[ \sqrt[3]{(d_p^{\ 3} + d_1^{\ 3})} - d_p \right] \tag{8}$$

$$CoagSrc = 0.5 \iint_{d_{p,l}^{\ 3} \le d_i^{\ 3} + d_j^{\ 3}}^{d_i^{\ 3} + d_j^{\ 3} < d_{p,u}^{\ 3}} \beta_{i,j} n_i n_j \cdot dlog d_i \cdot dlog d_j \tag{9}$$

where $d_{p,u}$ (nm) and $d_{p,l}$ (nm) are the upper and lower size limits of a given size bin; $d_p$ (nm) is the representative diameter (usually the geometric mean diameter) of this size bin; CoagS is the coagulation sink (s$^{-1}$) for $d_p$; CoagSrc is the coagulation source term (cm$^{-3}$·s$^{-1}$) for the given size bin; $N_{[dp,l\ dp,u]}$ is the measured concentration (cm$^{-3}$) of the size bin $[d_{p,l}, d_{p,u}]$ at $t_p$; $d_1$ is

the diameter (nm) of the condensing vapor; and $n_i$ is the aerosol size distribution function ($dN_i/dlog d_i$, cm$^{-3}$, where $N_i$ is the cumulative distribution) for the given size $d_i$.

The derivation for Eq. 6 is detailed in sections 4.1 and 4.2. The new and conventional appearance time methods are tested using a discrete-sectional model in section 4.3 and a measured atmospheric NPF event in section 4.4.

## 3 Methods

The proposed correction formula for influences of coagulation on the appearance time method (Eqs. 6 and 7) are derived in sections 4.1, 4.2. To test the validity of these corrections, numerical models are used to simulate an evolving aerosol size distribution and provide the theoretical growth rate according to the input monomer concentration. The growth rate is retrieved from the simulated aerosol size distribution using the conventional and corrected appearance time methods and then compared to the theoretical growth rate. Since the correction formulae are proposed only for the influence of coagulation, the influences of evaporation and varying vapor concentration on the retrieved growth rate with the appearance time method is discussed in section 4.3. After that, the conventional and corrected appearance time methods are applied to a new particle formation event measured in urban Beijing. The impact of coagulation on the appearance time of new particle and hence growth rates are indicated by the differences between the growth rates retrieved from the measured aerosol size distributions using the conventional and corrected appearance time methods.

### 3.1 Numerical models

A discrete aerosol model and a discrete-sectional aerosol model based on aerosol dynamics were used to provide an evolving aerosol size distribution and hence to test the conventional and corrected appearance time methods. The discrete model assumed that new particle formation is driven by the nucleation and condensation of a certain single-component condensing vapor ($H_2SO_4$). The vapor concentration was set as a constant. Condensation, coagulation, external loss, and evaporation were considered in this discrete model. The concentrations of particles up to the size of 100 vapor molecules (~3.5 nm) were numerically solved using the Julia programming language. The theoretical condensation and coagulation growth rates were calculated using Eqs. 1 and 2.

The discrete-sectional model was composed of 30 discrete bins (up to 2.2 nm) and 400 sectional bins (up to 230 nm). The vapor concentration was assumed to follow a normal distribution to simulate its diurnal variation in the real atmosphere. A growth enhancement factor as a function a particle size (Kuang et al., 2010) was used to account for the condensation of multiple vapors. A certain concentration of 100 nm particles was used as background particles and their concentration and size were kept constant during each simulation. The discrete-section model was coded in Matlab and it is detailed in Li and Cai (2020).

The simulation conditions for Figs. 1 and 3-7 are summarized in Table 2. The simulations with varying vapor concentration are summarized in Table A1. Note that the aerosol dynamic models are only used to provide a benchmark to compare the conventional and corrected appearance time methods in this study.

**Table 2**

**3.2 Measurements**

The NPF event measured on Feb. 24[th], 2018, in urban Beijing was used to test the influences of coagulation on the appearance time method. During 8:00 – 16:00 on this NPF day, the mean temperature, relative humidity, and wind speed are 1.3 °C, 22%, and 1.4 m s$^{-1}$, respectively.

The measurement site locates on the west campus of the Beijing University of Chemical Technology, which is close to

the west 3rd-ring road of Beijing. The aerosol size distributions were measured using a homemade particle size distribution system (PSD, Liu et al., 2016) and a homemade diethylene glycol scanning mobility particle spectrometer (Jiang et al., 2011a; DEG-SMPS, Cai et al., 2017) equipped with a core sampling apparatus (Fu et al., 2019). The sulfuric acid monomer and dimer concentrations were measured using a long chemical ionization time-of-flight mass spectrometers (ToF-CIMS, Aerodyne Research Inc., Jokinen et al., 2012). The meteorological data were measured using a local weather station (Vaisala, AWS310).

More details on this measurement site and the instruments have been introduced elsewhere (Deng et al., 2020b).

**4 Results and discussion**

**4.1 Appearance time method under ideal conditions**

Prior to investigating the impacts of coagulation on the appearance time method, we briefly illustrate the principle of the appearance time method. He et al. (2020) recently demonstrated that the appearance time method can retrieve the condensation

growth rate under ideal conditions. The ideal conditions are:

- The vapor concentration is constant;
- The initial concentrations of new particles are equal to zero;
- Condensation is the only cause of the change in particle concentrations, i.e., there is no coagulation, evaporation, external loss, etc.;

- The condensation rate (i.e., coagulation rate between vapor and particles) is independent of particle diameter.

It can be seen that none of these ideal conditions is consistent with real atmospheric environments. Since the conventional appearance time method is derived based on these conditions, violating them may cause biases in the appearance time method. We will first show brief derivations of the conventional appearance time and then discuss the correction for the influences of coagulation and other remaining potential uncertainties of the corrected appearance time method.

Under the above ideal conditions, the population balance equation for a particle containing i molecules is:

$$\frac{dN_i}{dt} = -\beta N_1 N_i + \beta N_1 N_{i-1} \ (i > 2) \tag{10}$$

where $N_1$, $N_{i-1}$, and $N_i$ are the concentrations (cm$^{-3}$) of the condensing vapor and particles containing i and i-1 monomer molecules, respectively; $t$ is time (s); $\beta$ is the coagulation coefficient (cm$^3 \cdot$s$^{-1}$) between a vapor molecule and any particle, which is assumed to be independent of the particle size in Eq. 10. For the case i = 2, the last term in Eq. 10 should be modified as $0.5\beta N_1^2$.

Solving the differential equations in Eq. 10 yields the analytical solution for $N_i$:

$$N_i(t) = N_{i,\infty} \times \left[1 - e^{-\beta N_1 t} \sum_{k=0}^{i-2} \frac{(\beta N_1 t)^k}{k!}\right] \tag{11}$$

where $N_{i,\infty}$ is the concentration limit (cm$^{-3}$) of $N_i$ when $t$ approaches infinite (d$N_i$/d$t$ = 0) and it is equal to $0.5\beta N_1^2$ (for i > 1) under these ideal conditions.

Figure 1a shows the concentrations of $N_i$ normalized by dividing by their corresponding $N_{i,\infty}$. It can be seen that the distance between two adjacent concentration curves is approximately a constant though these curves are not parallel. Hence, the appearance time method takes the moment that $N_i$ reaches a certain percentage of its maximum value ($N_{i,\infty}$) as its representative time. Previous studies indicate that the 50% size-resolved appearance time method which chooses the certain percent as 50% is more robust against non-ideal conditions compared to using the criterion of other percent values (Lehtipalo et al., 2014). As shown in Fig. 2, an approximate solution of $t$ for $N_i(t) = 0.5N_{i,\infty}$ (referred to as $t_i$) is:

$$t_i \approx \frac{\ln 2 + i}{\beta N_1} \tag{12}$$

Equation 12 indicates that the slope of particle size in terms of its molecule number versus its appearance time is approximately equal to its condensation growth rate, i.e.,

$$\frac{\Delta i}{\Delta t} = \frac{i - (i-1)}{t_i - t_{i-1}} \approx \beta N_1 = GR_{cond,n} \tag{13}$$

where $GR_{cond,n}$ is the condensation growth rate in terms of the molecule number (s$^{-1}$). The relationship between $GR_{cond,n}$ herein and $GR_{cond}$ in Eq.1 (which is defined with respect to particle diameter) is $GR_{cond} = GR_{cond,n} \times \Delta d_i$, where $\Delta d_i$ (nm) is the increase of $d_i$ due to the condensation of one vapor molecule.

**Figure 1**

According to the derivations above, the 50% size-resolved appearance time (referred to as 50% appearance time for short) method can retrieve particle growth rate under the given ideal conditions. The slope of particle size versus the appearance time is approximately equal to the condensation growth rate. That is, this slope is mainly determined by condensation growth under these ideal conditions. Note that Eq. 13 is only valid for the 50% appearance time whereas other thresholds to determine the appearance time may cause systematic bias. This bias comes from the non-parallelism of particle concentration curves (see Fig. 1). As shown in Fig. 2, in this test case, the 5% appearance time method overestimates the growth rate by 15% and the 95% appearance time method underestimates the growth rate by 12%. It should be clarified that since these biases are not huge,

it is acceptable to use other thresholds instead of 50% to reduce the impact of measurement uncertainties in practical applications.

Particle evaporation is assumed to be negligible in the above derivations, yet Fig. 2 indicates that size-independent evaporation does not significantly impact the validity of the 50% appearance time method. Assuming a size-independent evaporation rate, $E$ (s$^{-1}$), $t_i$ is approximately equal to $(\ln2+i)/(\beta N_1-E)$, which is bigger than that without evaporation. Meanwhile, considering evaporation, the net condensation growth rate (in the Lagrangian specification) is equal to the vapor condensation rate subtracted by particle evaporation rate, i.e., $\beta N_1-E$. That is, the increase in appearance time agrees with the decrease in the net condensation growth rate. In practice, particle evaporation rate is usually size dependent due to the significant Kelvin effect and it impacts $N_{i,\infty}$ in addition to the net condensation growth rate. It will be shown below in section 4.3 that for particles close to the critical size (corresponding to $\beta N_1-E = 0$), evaporation may cause a substantial bias in the net condensation growth rate retrieved using the appearance time method.

**Figure 2**

Note that the equality between the slope of particle size versus the 50% appearance time and the net condensation growth rate holds only under the above ideal conditions. The following derivations and results in section 4.2 will show how the slope is affected by coagulation while maintaining the same condensation growth rate.

### 4.2 The impacts of coagulation and their corrections

We first show the impact of an external sink on the appearance time method. The external sink is herein referred to as the sink due to coagulation with background particles, wall loss, dilution, transport, etc. For the convenience of comparison with Fig. 1a, the external sink is assumed to be temporally independent of particle diameter. The impact of its size dependency will be discussed later. Considering the constant external sink, the population balance equation for $N_i$ is:

$$\frac{dN_i}{dt} = -\beta N_1 N_i - ES N_i + \beta N_1 N_{i-1} \ (i > 2) \tag{14}$$

where ES is the external sink (s$^{-1}$) and other variables have been introduced in Eq. 10.

Similarly to Eqs. 11 and 12, the approximate solutions for $N_i$ and its corresponding appearance time ($t_i$) are:

$$N_i(t) = N_{i\infty} \times \left[1 - e^{-(\beta N_1 + ES)t} \sum_{k=0}^{i-2} \frac{[(\beta N_1 + ES)t]^k}{k!}\right] \tag{15}$$

$$t_i \approx \frac{\ln2 + i}{\beta N_1 + ES} \tag{16}$$

Equations 15 and 16 indicate that the impact of ES to $t_i$ is mathematically equivalent to vapor condensation ($\beta N_1$). In the presence of a non-negligible external sink, the particle concentration will approach its limit faster than the scenario without external sink (Fig. 1b). As a result, the slope of particle diameter versus appearance time is affected by both condensation growth and external sink.

Combining Eqs. 1, 13, and 16, the impact of external sink can be readily corrected. The correction formula is:

$$GR_{EScorr} = GR_{conv} - ES \cdot \left[ \sqrt[3]{(d_p{}^3 + d_1{}^3)} - d_p \right] \tag{17}$$

where $GR_{EScorr}$ is the growth rate (nm·s$^{-1}$) after correcting the external sink, $GR_{conv}$ is the growth rate (nm·s$^{-1}$) retrieved by the conventional appearance time method, and $d_1$ is the diameter (nm) of the condensing vapor.

As shown in Fig. 3a, with a vapor concentration of $5 \times 10^6$ cm$^{-3}$ and an external sink ranging from $1 \times 10^{-3}$ to $5 \times 10^{-3}$ s$^{-1}$, the conventional appearance time method overestimates the condensation growth rate substantially. Such an overestimation caused by mistaking the external sink for condensation growth was also reported in previous studies (Olenius et al., 2014; Li and McMurry, 2018). In contrast, the corrected growth rate agrees well with the theoretical condensation growth rate.

Practically, the coagulation coefficient ($\beta$) and ES are functions of the particle diameter. For the convenience of illustration, we use the size-dependent coagulation sink, CoagS, as an example to represent the total particle sink due to coagulation, wall loss, dilution, and transport. Similarly to Eq. 15, the approximate analytical solution for $N_i$ with the size-dependent $\beta$ and CoagS is:

$$N_i(t) \approx N_{i\infty} \times \left\{ 1 - e^{-(\beta_{1,i}N_1 + CoagS_i)t} \sum_{k=0}^{i-2} \left[ \frac{t^k}{k!} \prod_{g=i-k+1}^{i} (\beta_{1,g}N_1 + CoagS_g) \right] \right\} \tag{18}$$

where $\beta_{1,i}$ (or $\beta_{1,g}$) is the coagulation coefficient between a vapor molecule and a particle containing i (or g) molecules (cm$^3$·s$^{-1}$); $CoagS_i$ (or $CoagS_g$) is the coagulation sink of particles containing i (or g) molecules (s$^{-1}$); and other variables have been introduced above. Correspondingly, the ES term in Eq. 17 should be replaced with $CoagS_i$ to correct the impact of the size-dependent coagulation sink. When deriving Eq. 18, it is assumed that $\beta_{1,i}N_1 + CoagS_i$ is close to $\beta_{1,i-1}N_1 + CoagS_{i-1}$. This approximation is reasonable because both $\beta_{1,i}$ and $CoagS_i$ change gradually with the particle size, yet it introduces minor systematic biases in $N_i$ and its corresponding appearance time.

As shown in Fig. 3b, when $\beta_{1,i}$ and $CoagS_i$ are size-dependent, the corrected appearance time method is still able to reproduce the condensation growth rate. It is assumed that the $CoagS_i$ in Fig. 3b is contributed by only the large background particles. Hence, the $CoagS_i$ in Fig. 3b is estimated from the condensation sink (CS) using an empirical formula (Eq. 8 in Lehtinen et al., 2007), where CS indicates the condensation loss rate of the vapor.

**Figure 3**

In addition to the coagulation with another background particle, the coagulation between two new particles also contributes to the CoagS of both these two particles. As explained in section 2.2, no matter how small the coagulating particle is, the coagulation between a given particle and any other particle should be accounted for in CoagS. This is because the appearance time method is derived in the Eulerian specification and the CoagS is defined for a certain particle diameter rather than for a certain particle. In contrast, when focusing on the survival probability of new particles (Weber et al., 1997; Lehtinen et al., 2007), CoagS should be calculated in the Lagrangian specification, i.e., only the coagulation with a larger particle that causes particle loss should be accounted for. To emphasize the difference between the two definitions of CoagS, the corrected growth rates using the total (Eulerian) CoagS and the background (Lagrangian) CoagS are compared in Fig. 4. Particle source

due to coagulation is not considered in this comparison. A constant concentration of 100 nm particles used as the background particles. The background CoagS refers to the sink due to coagulation with all larger particles, including both the background particles and new particles. The total CoagS is calculated using Eq. 3. Note that due to the contribution of new particles, the total CoagS does not follow a simple decreasing trend with the increasing particle diameter (see Fig. B1 in Cai and Jiang, 2017). Hence, the empirical formula (Lehtinen et al., 2007) to generate a size-dependent CoagS in Fig. 4b should not be used for the total CoagS.

As shown in Fig. 4, the condensation growth rate after correcting the background CoagS is still overestimated. In contrast, the growth rate after correcting the total CoagS agrees with the theoretical growth rate for particles larger than 1.3 nm. For sub-1.3 nm particles, the systematic bias of the growth rate after correcting the total CoagS is mainly caused by the violation of the assumption that $\beta_{1,i}N_1+\text{CoagS}_i$ is close to $\beta_{1,i-1}N_1+\text{CoagS}_{i-1}$. The size dependency of particle coagulation coefficient under the influence of new particle coagulation increases with decreasing particle size. For instance, under the test conditions, $(\beta_{1,4}N_1+\text{CoagS}_4)/(\beta_{1,3}N_1+\text{CoagS}_3) = 1.12$ while $(\beta_{1,50}N_1+\text{CoagS}_{50})/(\beta_{1,49}N_1+\text{CoagS}_{49}) = 1.01$. As a result, the corrected appearance time method still overestimates the growth rate for sub-1.3 nm particles.

**Figure 4**

In addition to CoagS, the impact of coagulation source on the appearance time also needs correction. Adding the coagulation source term to the population balance equation of $N_i$ yields:

$$\frac{dN_i}{dt} = -\beta_{1,i}N_1N_i - \text{CoagS}_iN_i + \beta_{1,i-1}N_1N_{i-1} + \text{CoagSrc}_i \ (i > 2) \tag{19}$$

$$\text{CoagSrc}_i = 0.5\sum_{k=2}^{i-2}\beta_{k,i-k}N_kN_{i-k} \tag{20}$$

where $\text{CoagS}_i$ is the coagulation sink (s$^{-1}$) corresponding to $N_i$, $\text{CoagSrc}_i$ is the coagulation source term (cm$^{-3}\cdot$s$^{-1}$) corresponding to $N_i$, and other variables have been introduced above.

Since $\text{CoagSrc}_i$ is determined by the concentrations of all particles containing 2 to i-2 molecules, it is difficult to obtain an analytical solution of Eq. 19 without approximation. Here we provide an approximation method to correct the impact of coagulation source to the appearance time. Compared to the scenario that there is no coagulation source term in the balance equation (Eq. 19), the coagulation source has three impacts on particle size distribution and particle growth: 1) the coagulation with a smaller particle contributes to particle growth; 2) the coagulation source increases the maximum particle concentrations; 3) the coagulation source shortens the time for particles to reach their maximum concentration.

These three impacts of coagulation source are accounted for in the corrected appearance time method (Eqs. 6 and 7). Impact 1) is corrected using Eq. 7. To correct impacts 2) and 3), we simply assume that $\text{CoagSrc}_i$ is a constant during the increasing period of $N_i$ and use the following formulae (see the SI for its derivation) to estimate the growth rate. The correction formula has been given in Eq. 6.

The corrected appearance time method was tested under various modeling conditions. As the example in Fig. 5 indicates, the growth rates estimated using the corrected appearance time method agrees with the theoretical growth rates. Equation 5 is

able to retrieve the growth rate of sub-3 nm particles unless when coagulation source is a governing reason for the change of particle concentration, i.e., CoagSrc/$2N_p$ is comparable or larger than $\beta_{1,p}N_1$. Under these conditions, CoagSrc may not be a constant and, hence, the approximation of CoagSrc/$2N_p$ may cause bias. Fortunately, CoagSrc usually decreases with the increasing particle size due to the decreasing particle concentration. Furthermore, it will be shown in section 4.4 that CoagSrc

does not have a major impact on the apparent growth rate of sub-10 nm particles even during an intensive atmospheric NPF event in urban Beijing. Hence, we consider Eq. 6 as a rough but sufficient formula to correct the impact of coagulation on the appearance time for most atmospheric NPF events.

**Figure 5**

## 4.3 Uncertainties of the appearance time method

In the above analysis, the correction for the influences of coagulation on the appearance time method was validated by derivations and simulations. However, there are still potential uncertainties in the corrected appearance time method because of other uncorrected influences. This section will discuss the uncertainties caused by vapor evaporation and a varying vapor concentration.

Due to the uncorrected vapor evaporation, the appearance time method may overestimate the growth rate for when the

net condensation rate in the Lagrangian specification ($\beta N_1 - E$) is close to zero or negative. As shown in Fig. 6a, a size-dependent evaporation rate is assumed for the clusters and particles. The evaporation rate of a dimer (i.e., cluster containing two vapor molecules) is assumed to be ~0.2 s$^{-1}$ according to the stability of $H_2SO_4$-$NH_3$ clusters reported in Myllys et al. (2019). The evaporation rate of particles containing more than three vapor molecules is estimated using the Kelvin equation with arbitrarily assumed vapor saturation pressure. With these assumptions of vapor evaporation, the evaporation rate ($E$) of a single cluster

exceeds its monomer association rate ($\beta N_1$) in the sub-1.5 nm size range (Fig. 6a). The influence of coagulation source is not accounted for in this test yet its influences can be corrected according to the above discussions in section 4.2. As shown in Fig. 6b, the conventional and corrected appearance time method overestimates the net condensation growth rate of sub-2 nm particles. This bias is mainly caused by the influence of the size-dependent evaporation on $N_{i,\infty}$. Assuming a prior knowledge of the size-dependent evaporation rate, the net condensation growth rate retrieved using the appearance time method can be

corrected using Eq. 21:

$$GR_{corr,i} \; = \; GR_{conv,i} \times \frac{\beta_{1,i}N_1 - E_{i+1}}{\beta_{1,i}N_1 - E_{i+1}N_{i,50}/N_{i+1,50} + CoagS_i} \tag{21}$$

where the subscripts i and i+1 indicates the number of molecules contained in the clusters; $N_{i,50}$ and $N_{i+1,50}$ are the concentrations of $N_i$ and $N_{i+1}$, respectively, at the 50% appearance time of $N_i$; and other variables has been declared above. $N_{i,50}$ and $N_{i+1,50}$ can be approximated by $N_{i,\infty}$ and $N_{i+1,\infty}$, respectively. The last term in Eq. 21 corrects the influence of evaporation on $N_{i,\infty}$ and the source of $N_i$.

**Figure 6**

As shown in Fig. 6b, the appearance time method corrected using Eq. 21 follows the theoretical net condensation growth rate. However, since Eq. 21 requires prior knowledge of the size-dependent evaporation rate, it is difficult to use this a correction for an atmospheric NPF event. Hence, one should note the uncertainties of the appearance time method for particles in the neighbor of the critical size.

5 Note that the above discussions are for the case that nucleation and growth are driven by a single volatile vapor. The condensation of a volatile vapor onto a non-volatile particle may not introduce a significant bias to the growth rate estimated using the appearance time method, as shown in Fig. S1 in the supplementary information (SI). This is because with an existing non-volatile vapor, $N_{i,\infty}$ may not be significantly affected by the evaporation of other volatile vapors.

The constant vapor concentration assumption may be valid for some chamber experiments; however, the vapor 10 concentration usually follows a diurnal pattern in the real atmosphere. The varying vapor concentration may impact the appearance time and hence the retrieved apparent growth rate. As reported in previous studies (Lehtipalo et al., 2014; Olenius et al., 2014), the retrieved appearance time is sensitive to the variation of vapor concentration. In the presence of coagulation, it is difficult to correct the impact of the varying vapor concentration. Herein, we use the discrete-sectional model to test the uncertainties of the corrected appearance time method under a varying vapor concentration. The vapor concentration is 15 assumed to follow a normal distribution. The condensation sink ($10^{-3}$ – $10^{-2}$ $s^{-1}$) is contributed simultaneously by a certain number concentration ($7.2\times10^2$ – $7.2\times10^3$ $cm^{-3}$) of 100 nm background particles and the new particles. The growth rate is firstly estimated using the 50% size-resolved appearance time method and then corrected using Eq. 8. The 50% appearance time is herein calculated using the maximum size-resolved particle concentration because $N_{i,\infty}$ is not available, and this approximation introduces biases to the retrieved growth rate. Since the vapor concentration varies with time, the retrieved growth rate 20 characterizes particle growth at both different diameters and different time instead of the size-dependent growth at a certain moment. To keep in accordance with the appearance time method, the theoretical condensation and coagulation growth rates of each $d_p$ are calculated at its corresponding $t_p$.

In general, neglecting the variation of the vapor concentration introduces biases to the appearance time method. As the example shown in Fig. 7 (test No. 8 in Table) indicates, the deviation between the corrected particle growth rate and the 25 theoretical growth rate is smaller than the deviation between the conventional growth rate and the theoretical growth rate. However, for particles larger than ~5 nm and smaller than ~2 nm, the appearance time method overestimates particle growth rate even after correcting the impact of coagulation in the test case. These overestimations are caused by approximating the influence of coagulation source on particle growth with the coagulation source term in Eq. 6. As summarized in Table A1, the relative discrepancy depends on the exact conditions. The average discrepancy of the corrected appearance time method for 30 1.5-3 nm particles ranges from 6% to 44% in the test conditions, which is smaller than that of the conventional method. The maximum size-dependent discrepancy of the corrected appearance time method reaches 150% under the test conditions.

Although it is difficult to correct the bias due to the varying vapor concentration, one can try to avoid large uncertainties because the bias seems to follow a certain pattern. Compared to the scenario of an increasing vapor concentration, it is found that the discrepancy between the real and retrieved growth rates are usually larger after the peak time of the vapor concentration.

As shown in Fig. 7, the appearance time of ~4.9 nm particles is 12:00 and a substantial discrepancy between the theoretical and retrieved growth rate is observed for particles larger than 4.9 nm. Fortunately, during a typical atmospheric NPF event, new particles usually grow large before the vapor concentration starts to decrease. To reduce this systematic bias due to a decreasing vapor concentration, we suggest using the other methods, e.g., the representing diameter method to estimate particle

growth rate when the vapor concentration decreases.

**Figure 7**

For particles close to the size of vapor molecules (sub-2 nm in these tests), the appearance time usually convolves other information (e.g., the varying vapor concentration and the size-dependent coagulation coefficient) in addition to particle growth. Figure S2 shows that with larger vapor molecules, the size range for the discrepancy between the theoretical and retrieved

growth rate shifts towards larger diameters. Considering the influences of vapor evaporation and varying vapor concentration on the appearance time method, one should be cautious about the size-resolved growth rate for particles close to the size of vapor molecules.

CoagS is assumed to be independent of time in the above discussions, whereas it may vary significantly during an NPF event in the atmosphere. The varying CoagS influences $N_{i,\infty}$ and hence the appearance time. Figure S3 shows that a bias of the

growth rate retrieved using the appearance time method caused by a varying CoagS.

In addition to vapor evaporation and the variation of vapor concentration, there may be other limitations for determining the appearance time in the atmosphere. Differently from controlled chamber studies (Dada et al., 2020), the uncertainties in atmospheric measurements pose challenges to growth rate estimation. These uncertainties come from instrumental biases, atmospheric turbulence, and the omitted contributions from transport, mixing, and emissions to the measured aerosol size

distribution. Since the growth rate is calculated using a differential formula (Eq. 5), it is usually more sensitive to uncertainties than a physical quantity calculated using an integral formula (e.g., CoagS). For instance, the appearance time as a function of particle diameter in Fig. 7 had to be smoothed before calculating the growth rate using Eq. 5; otherwise, the calculated growth rate at some certain size bins would be negative. The applications of other methods to estimate the particle growth rate face the same challenge. As discussed in the Introduction, the appearance time method is used to estimate new aerosol growth rate

because other methods sometimes cannot report a growth rate in the concerned size range. Hence, further investigations concerning the uncertainties are needed for a better estimation of the growth rate in the atmosphere.

### 4.4 Application in atmospheric measurements

A typical intense NPF event measured in urban Beijing is used to test the correction for the influences of coagulation on the appearance time method. During the event, the peak sulfuric acid concentration was ~$6\times10^6$ cm$^{-3}$ and the average CS for

sulfuric acid was 0.024 s$^{-1}$. The theoretical condensation and coagulation growth rates are calculated using the measured sulfuric acid concentration and aerosol size distribution, respectively. Particle growth due to the uptake of sulfuric acid dimers is herein accounted for as the condensation growth. Note that the sum of theoretical condensation and coagulation growth rate is not necessarily equal to the theoretical total growth rate for the measured NPF event. This is because only the condensation

of sulfuric acid is considered whereas other vapors may also contribute to new particle growth. The enhancement due to Van der Waals force is considered when calculating the coagulation coefficient (Alam, 1987; Chan and Mozurkewich, 2001; Stolzenburg et al., 2019). The appearance time retrieved from the measured aerosol size distributions was smoothed before estimating the particle growth rate. In addition to the example given in Fig. 8, the average growth rates of 1.5-3 nm particles measured in urban Beijing retrieved using the conventional and corrected appearance time methods were reported in Deng et al. (2020a).

As shown in Fig. 8, the measured $H_2SO_4$ concentration started to increase around 7:00 and its 50% appearance time was 7:30. Because $SO_2$ concentration declined between 8:00 and 14:00, the peak concentration of $H_2SO_4$ was observed at 8:00 rather than at noon. Shortly after the increase in $H_2SO_4$ concentration, new particles down to the cluster size (~1 nm in geometric diameter) was observed. The high concentration of sub-2 nm aerosol during 7:00 - 15:00 and the continuous growth pattern of new particles from 1.5 nm to 10 nm indicates that the observed new particle formation event was a typical regional event (Kulmala et al., 2012), though transport, mixing, and emissions might influence the observed aerosol size distributions.

The impacts of particle coagulation are non-negligible compared to particle growth and the growth rate calculation in urban Beijing. On one hand, the conventional appearance time method overestimates the particle growth rate for sub-3 nm particles in urban Beijing due to the impact of CoagS. The deviation between the conventional and corrected growth rate decreases with the increasing diameter because CoagS decreases with particle diameter. As illustrated above, the correction for CoagSrc is only an approximation rather than obtained based on solid derivations. However, the negligible impact of CoagSrc on the measured growth rate in urban Beijing indicates that this approximation does not cause significant bias. Different from coagulation growth which is weighted by particle size, the CoagSrc of $d_p$ is only determined by the number concentrations of particles smaller than $d_p$ (and their coagulation coefficient). Even under such an intense NPF event (with the maximum formation rate exceeding 200 $cm^{-3} \cdot s^{-1}$), the new particle concentration is usually much smaller than the vapor concentration due to the high CoagS and possibly cluster evaporation. Hence, it is sometimes acceptable to neglect the CoagSrc/$N_p$ term in Eqs. 6 and 8 to facilitate calculation. On the other hand, the coagulation with smaller particles enhances particle growth and this enhancement increases with the increasing particle size. This emphasizes that during an intensive NPF event with a high new particle concentration, the condensation growth rate contributed by condensing vapors cannot be taken as the total growth rate that determines the survival probability of new particles.

**Figure 8**

The difference between the measured and theoretical growth rates in Fig. 8 also indicates the growth mechanism of new particles. Considering the uncertainties of the appearance time, the sum of condensation and coagulation flux of sulfuric acid molecules and clusters is approximately equal to the measured particle growth rate for ~3 nm particles, which indicates that sulfuric acid is a governing species that contribute to the initial growth of sub-3 nm particles during the test event. The deviation between the measured growth and theoretical growth for particles larger than ~3 nm indicates that there are other chemical species in addition to sulfuric acid (and the bases to neutralize it) contributing to particle growth. Note that the above discussion

is only based on a single case study. Hence, further investigations based on long-term measurements are needed to reveal the growth mechanism in the polluted atmospheric environment.

Summarizing all the analysis above, the growth rate retrieved using the conventional appearance time method may be systematically overestimated due to the impact of coagulation, especially for intensive NPF events in polluted environments. Such an overestimation may be significant for sub-3 nm particles because CoagS increases with the decreasing particle size. In addition, the coagulation growth rate also needs to be corrected before investigating the condensation growth mechanism. For example, in the test case shown in Fig. 8, the retrieved condensation growth rate of 1.5 nm particles using the conventional appearance time method without correcting the impact of CoagS and the coagulation growth rate is overestimated by 80%. Figure 8 also indicates that the impact of CoagS may be negligible for larger particles and clean environments (see also Fig. S4). However, external sinks (e.g., dilution) may also cause an overestimation of the growth rate retrieved using the appearance time method if they are not properly corrected.

## 5 Conclusions

The impact of coagulation on the particle growth rate retrieved using the appearance time method was investigated based on theoretical derivations and aerosol dynamics modeling. It was found that the often-used 50% size-resolved appearance time method can reproduce the condensation growth rate only under the idealized condition without particle coagulation. When using the appearance time method in the real world, coagulation sink, coagulation source, and coagulation growth need to be considered. Equations 5-9 provide a method in both discrete and sectional forms to correct the impacts of coagulation sink and coagulation source to the appearance time method. The feasibility of the correction for the influences of coagulation was verified using discrete and discrete-sectional aerosol models. In addition, vapor evaporation and the variation of vapor concentration was found to impact the appearance time method. The average uncertainties of the corrected 1.5-3 nm particle growth rate for each NPF event were 6-44% in the test cases and the maximum size-dependent uncertainty was 150%. These uncertainties indicate that even after the correction for coagulation, one should be cautious about the appearance time method for particles close to the size of vapor molecules. Further, the growth rate of vapors and clusters is recommended to be estimated based on cluster dynamics instead of their representative time.

A typical NPF event measured in urban Beijing was used to show the quantitative impacts of coagulation on the retrieved growth rate. The systematic bias of the conventional appearance time method was observed for sub-3 nm particles due to the uncorrected impact of the coagulation sink. Besides, coagulation growth was non-negligible compared to the growth due to sulfuric acid condensation, which emphasizes the importance to distinguish the condensation and total growth rates. During the test event, the apparent growth rate of 1.5 nm particles retrieved using the conventional method was 80% higher than the corrected condensation growth rate, whereas the corrected condensation growth rate was approximately equal to the theoretical growth rate contributed by sulfuric acid condensation.

## Appendix

**Table A1**

**Derivation of Eq. 1**

Consider a particle population with a uniform diameter of $d_p$ (nm) and a concentration of $N_0$. $N_0$ is assumed to be sufficiently large so that the stochastics in particle growth is negligible. At the initial moment $t_0$ (s), the mean particle diameter is $d_p$. During a short time interval $dt$ (s), $\beta_{1,p}N_1N_0dt$ particles collide with the condensing vapor with a diameter of $d_1$, where $\beta_{1,p}$ is the coagulation coefficient ($cm^3 \cdot s^{-1}$) and $N_1$ is the vapor concentration. Hence, the mean diameter ($\overline{d_p}$) weighted by particle number concentration at the moment $t_0+dt$ is:

$$\overline{d_p}(t_0 + dt) = \beta_{1,p}N_1dt\sqrt[3]{d_p{}^3 + d_1{}^3} + \left(1 - \beta_{1,p}N_1dt\right)d_p \tag{Eq. A1}$$

Comparing $\overline{d_p}(t_0)$ and $\overline{d_p}(t_0 + dt)$ yields the condensation growth rate:

$$\text{GR}_{\text{cond}} = \frac{\overline{d_p}(t_0 + dt) - \overline{d_p}(t_0)}{dt} = \beta_{1,p}N_1 \cdot \left[\sqrt[3]{\left(d_p{}^3 + d_1{}^3\right)} - d_p\right] \tag{Eq. A2}$$

The Taylor series of Eq. A2 is:

$$\text{GR}_{\text{cond}} = \frac{\beta_{1,p}N_1d_1{}^3}{3d_p{}^2} + o\left[\beta_{1,p}N_1d_p\left(\frac{d_1}{d_p}\right)^6\right] \tag{Eq. A3}$$

where $o\left[\beta_{1,p}N_1d_p\left(d_1/d_p\right)^6\right]$ is the Peano form of the remainder. The first term on the right-hand side of Eq. A3 is the formula for particle growth rate in the continuous form and the second term (the remainder) is the difference between the growth rate formula in the continuous and discrete forms (Olenius et al., 2018). When $d_p$ is sufficiently larger than $d_1$, Eq. A2 is reduced to $\beta_{1,p}N_1d_1{}^3/\left(3d_p{}^2\right)$.

*Author contributions*. RC, JK, JJ, and MK initialized the study. RC and CL developed the models. RC and X-CH derived the formulae. CD, YL, RY, CY, LW, and JJ performed the measurements and analyzed the data. RC did the simulation and wrote the manuscript with the help of other co-authors.

*Code availability*. The Julia code for the discrete model is available upon request. The Matlab code for the discrete-sectional model can be found via the link in Li and Cai (2020).

*Competing interests*. The authors declare that they have no conflict of interest.

*Acknowledgments*. Financial supports from the Academy of Finland project (332547, 1325656), UHEL 3-year grant (75284132), National Key R&D Program of China (2017YFC0209503), and Samsung $PM_{2.5}$ SRP are appreciated.

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

**Tables and Figures**

Table 1 The impacts of coagulation characterized in the Lagrangian and Eulerian specifications.

| | Coagulating with a smaller particle | Coagulating with a larger particle |
|---|---|---|
| Lagrangian: tracking individual particles | Coagulation growth (GR$_{coag}$) | Coagulation loss |
| Eulerian: tracking a given size bin | Coagulation sink (CoagS) for the current bin Coagulation source (CoagSrc) for the next bin | |

Table 2 The simulation conditions for Figs. 1 and 3-6. The symbol "√" indicates "yes" and the blank indicates "no".

| Figure No. | Size-dependent coagulation coefficient? | External sink? | Coagulation sink? | Coagulation source? | Vapor evaporation? | Constant vapor concentration? |
|---|---|---|---|---|---|---|
| 1a | | | | | | √ |
| 1b & 3a | | √ | | | | √ |
| 3b & 4 | √ | | √ | | | √ |
| 5 | √ | | √ | √ | | √ |
| 6 | √ | | √ | | √ | √ |
| 7 | √ | | √ | √ | | |

Table A1 The mean and maximum relative errors of the conventional and corrected appearance time methods for 1.5-3 nm particles. Conv. and corr. are short for the conventional and corrected methods, respectively. The vapor concentration is assumed to follow a normal distribution with a peak concentration of $N_{max}$ and a standard deviation of $\sigma_t$. Background CS
10  characterizes the concentration of 100 nm background particles. The errors are given in relative values. The results of No. 8 test is shown in Fig. 6.

| No. | $N_{max}$ (cm$^{-3}$) | $\sigma_t$ (h) | Background CS (s$^{-1}$) | Mean error conv. | Mean error corr. | Max. error corr. |
|---|---|---|---|---|---|---|
| 1 | $5.0\times10^6$ | 2 | $2\times10^{-3}$ | 35% | 8% | 19% |
| 2 | $2.0\times10^6$ | 2 | $2\times10^{-3}$ | 63% | 32% | 77% |
| 3 | $3.5\times10^6$ | 2 | $2\times10^{-3}$ | 30% | 6% | 18% |
| 4 | $8.0\times10^6$ | 2 | $2\times10^{-3}$ | 71% | 44% | 150% |
| 5 | $5.0\times10^6$ | 1 | $2\times10^{-3}$ | 38% | 11% | 40% |
| 6 | $5.0\times10^6$ | 3 | $2\times10^{-3}$ | 38% | 10% | 16% |
| 7 | $5.0\times10^6$ | 4 | $2\times10^{-3}$ | 39% | 11% | 17% |
| 8 | $5.0\times10^6$ | 2 | $1\times10^{-3}$ | 50% | 24% | 88% |
| 9 | $5.0\times10^6$ | 2 | $5\times10^{-3}$ | 21% | 20% | 66% |
| 10 | $5.0\times10^6$ | 2 | $1\times10^{-2}$ | 37% | 29% | 94% |

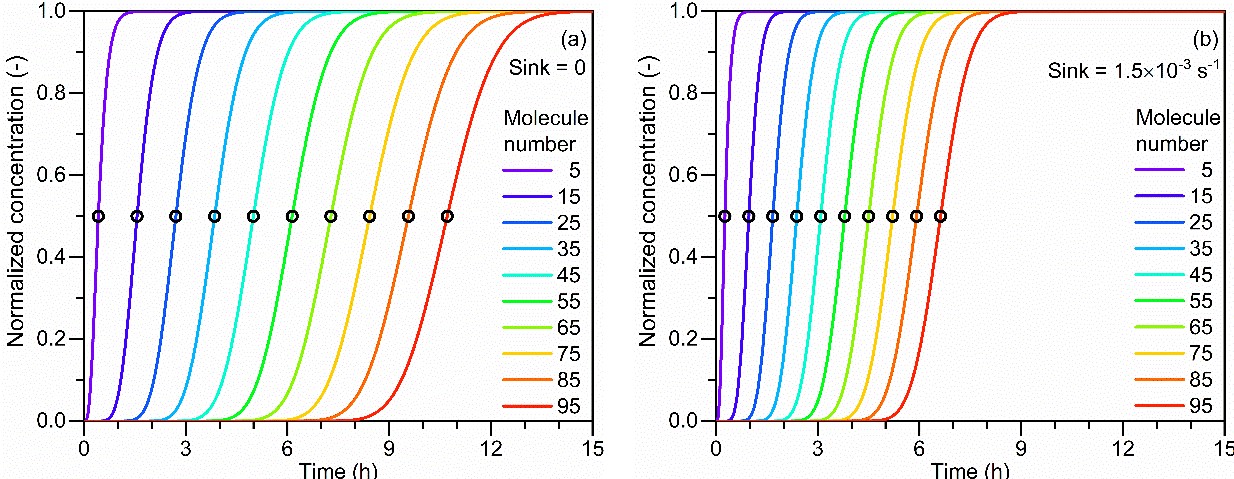

**Figure 1** The principle of the appearance time method and the impact of the external sink. (a) Normalized particle concentrations as a function of time. The concentrations are normalized by dividing their corresponding maximum concentrations. The number concentration of the condensation vapor is assumed to be constantly $5 \times 10^6$ cm$^{-3}$. Particle coagulation sink and other sinks are assumed to be negligible. Particle size is indicated by the molecule number contained in every single particle. The open scatters indicate the 50% appearance time corresponding to each particle size. (b) A constant external sink of $1.5 \times 10^{-3}$ s$^{-1}$ is considered and other simulation conditions are the same as a). Note that due to the assumption of the size-independent coagulation coefficient, the appearance time in this figure deviates from their typical values in real new particle formation events.

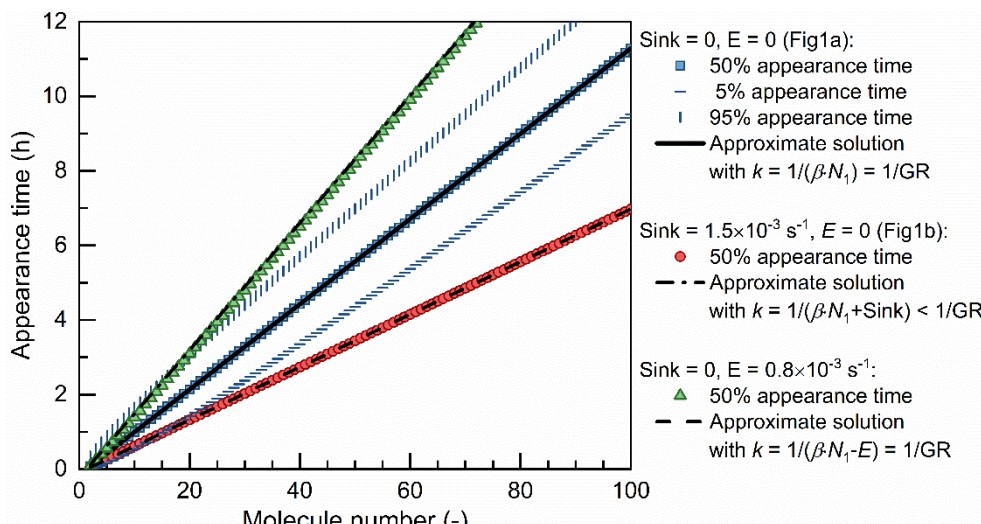

**Figure 2** The retrieved appearance time as a function of particle size. The particle size is characterized by the number of molecules contained in each single particle. The scatters are the appearance time retrieved from the simulated concentrations. The curves are the approximate solutions for the 50% appearance time, $t_i = k \times (i + \ln 2)$, where $k$ is the slope of the curve (see Eqs. 7 and 11). $\beta$ is the coagulation coefficient ($cm^3 \cdot s^{-1}$) between vapor and particles, $N_1$ is the vapor concentration ($5 \times 10^6$ cm$^{-3}$), and $E$ is the particle evaporate rate (s$^{-1}$). When sink $= 0$, the slope of the approximate solution is equal to the theoretical net condensation growth rate (GR, in terms of the molecule number contained in every single molecule), $\beta N_1 - E$. However, when sink $> 0$, the apparent growth rate, $\beta N_1 + Sink$, is higher than the theoretical condensation growth rate, $\beta N_1$.

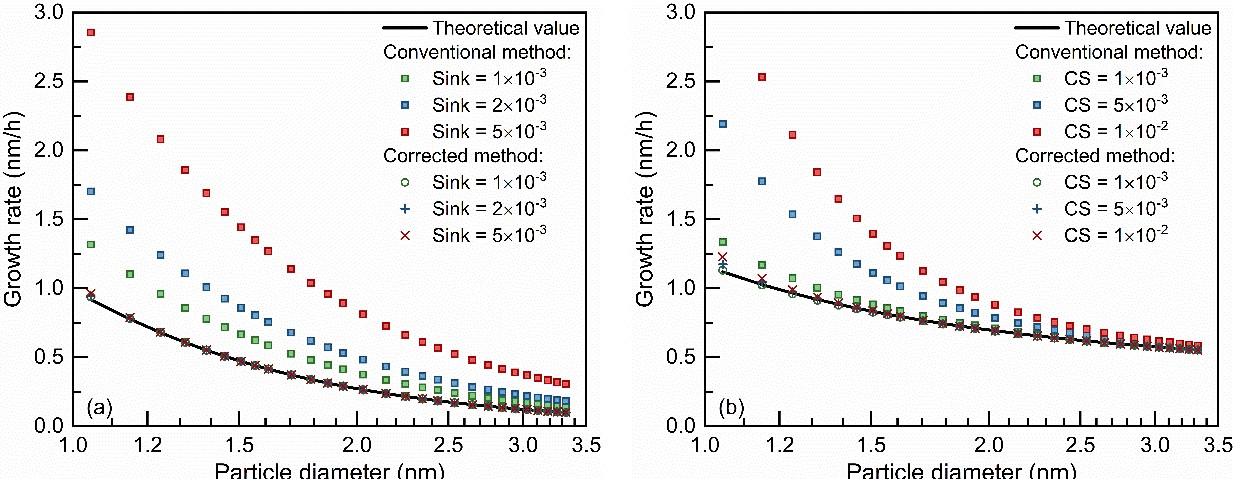

**Figure 3** The impact of sinks for the appearance time method and its correction. The theoretical curve is obtained using the condensation rate of the condensing vapor (Eq. 1) and the scatters are obtained using the conventional and corrected appearance time method. The sink is assumed to be independent and dependent of particle diameter in (a) and (b), respectively. The scatters
5  for the corrected method lie on top of each other. The size-dependent coagulation sink in (b) was estimated from the condensation sink (CS) shown in legend using an empirical formula (Lehtinen et al., 2007). The coagulation sink is taken as the input value of the model, hence, the validity of the empirical formula does not affect the accuracy of simulated size distribution or the growth rate. The minor discrepancy among the corrected growth rate comes from the size-dependent particle coagulation coefficient and coagulation sink.

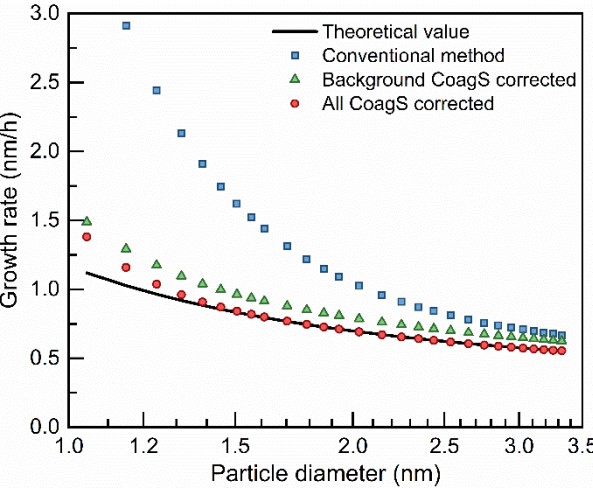

15  **Figure 4** The impact of coagulation sink (CoagS) due to colliding with a smaller particle to the appearance time method. Only the coagulation with a larger particle is accounted for in the correction using the background CoagS.

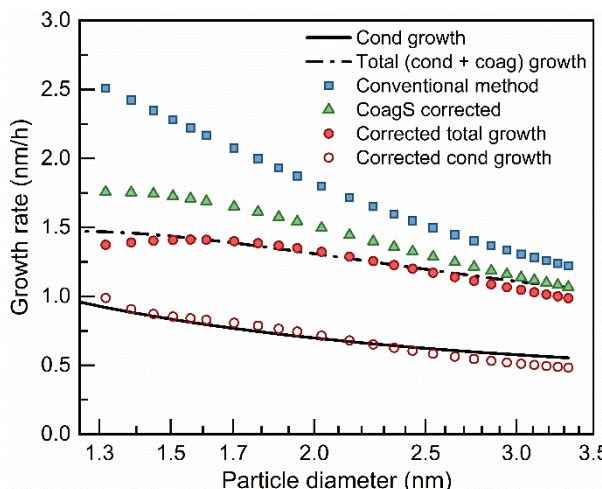

**Figure 5** The impact of coagulation sink (CoagS) and coagulation source (CoagSrc) to the appearance time method and the contribution of coagulation growth. Cond and coag are short for condensation and coagulation, respectively. Note that both the solid and dashed lines are theoretical growth rates and their difference is the coagulation growth. Similarly, both the open and filled circles are the measured growth rates after correction and their difference is equal to the coagulation growth rate.

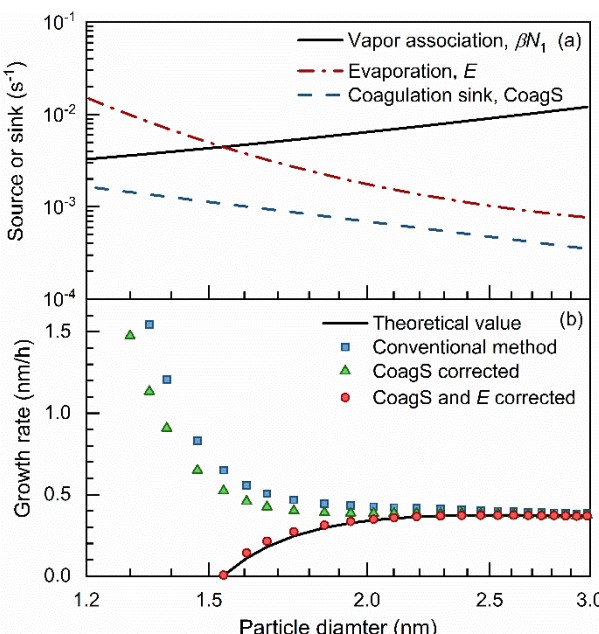

**Figure 6** The impact of vapor evaporation on the appearance time method. (a) The size-dependent vapor association rate ($\beta N_1$), evaporation rate ($E$), and coagulation sink (CoagS). (b) The theoretical net condensation grow rate and the growth rate retrieved using conventional and corrected appearance time methods. The theoretical growth rate is defined in the Lagrangian specification, i.e., it is calculated using $\beta N_1$-$E$.

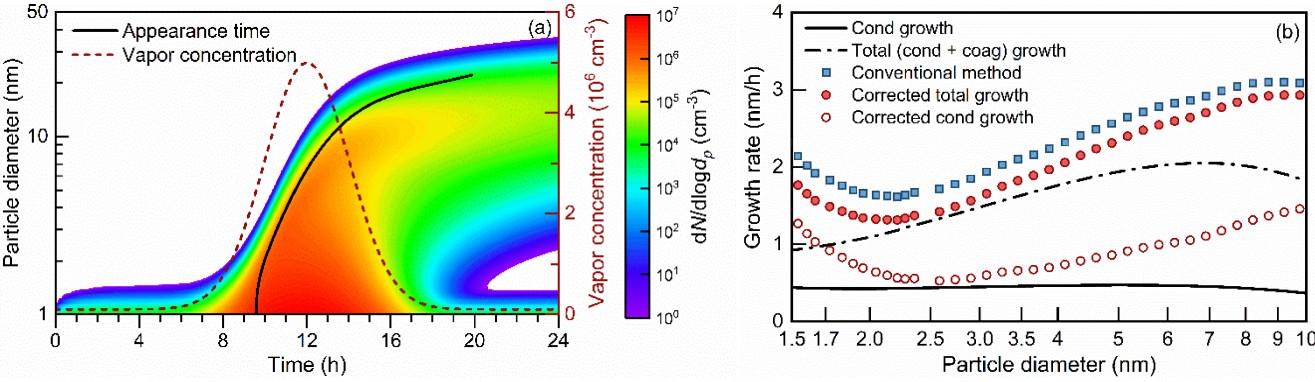

**Figure 7** The appearance time method under a varying vapor concentration. The test condition is summarized in Table A1, No. 8. (a) An NPF event simulated using a discrete-sectional aerosol dynamic model. The vapor concentration is assumed to follow a normal distribution (with a background value of $10^5$ cm$^{-3}$). The 100-nm background particles are not shown. The particle diameter as a function of the appearance time is shown in the solid line. (b) The theoretical and retrieved particle growth rates. Cond and coag are short for condensation and coagulation, respectively.

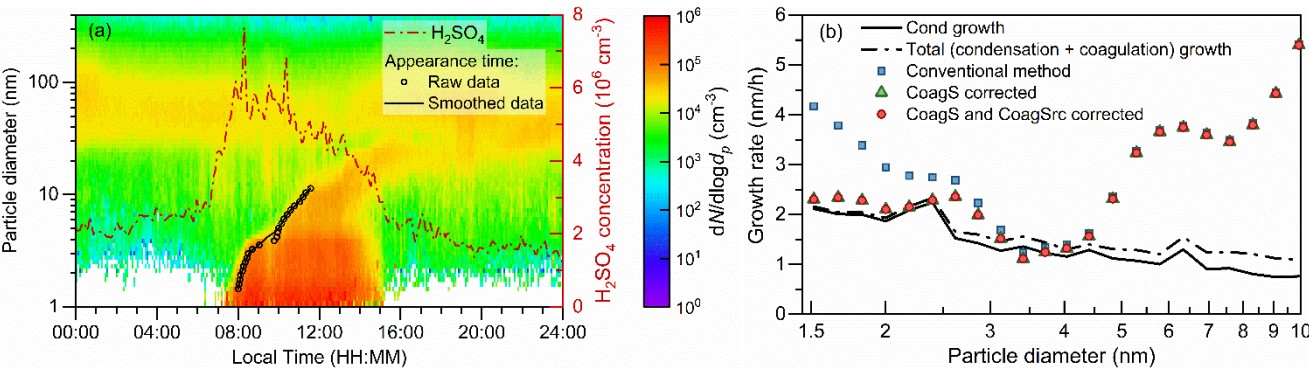

5    **Figure 8** A case study for the appearance time method in the real atmosphere. (a) Aerosol size distribution and $H_2SO_4$ concentration measured on an NPF day. The event was measured on Feb. 24th, 2018, in urban Beijing. (b) Measured growth rates using the conventional and corrected appearance time methods and the theoretical growth rate contributed by sulfuric acid condensation and particle coagulation. Note that the theoretical growth rate considers only the sulfuric acid condensation, hence, it may underestimate the overall condensation growth rate contributed by multiple condensing vapors.

