# Peer review of "Impacts of coagulation on the appearance time method for new particle growth rate evaluation and their corrections"

_Atmospheric Chemistry and Physics, 2020_

## Referee Comment (RC1) · Wolfgang Junkermann (Referee) · 26 Jul 2020

Summary

The manuscript describes a numerical set of formulas that is used in an aerosol model for the calculation of the growth of aerosol particles after nucleation, especially for the range of $\sim$ 1 nm to 3 nm. This formula set is used for a calculation of the appearance time (time after cluster formation) and later for estimation of the growth rate of the larger sizes aerosols > 4 nm although the main emphasis is on the smallest size ranges, where coagulation has a major effect. Model results based on these formulas are finally compared to a nucleation mode particle event in a relative highly polluted environment

in China, in the city of Beijing in a late winter situation.

General comments

The argumentation and description of the formulas seems plausible and is according to the authors in agreement with the theory. Unfortunately this 'reference' theory, which is several times cited in the manuscript, is neither described nor referenced. Also there is no information about this general 'theory' and the underlying nucleation scheme applied. There are several nucleation schemes, binary nucleation of sulfuric acid and water (neutral and ion-assisted), ternary involving sulfuric acid, water, and ammonia; with or without organic species and sulfuric acid or charged sulfuric acid-water-ammonia (Napari et al, 2002; Riccobono et al. 2014; Dunne et al. 2016). Otherwise, the results of the subsequent calculations are at least partially contradicting previous publications that are a base for many 'new particle formation' studies within the last decade.

That discrepancy is likely due to the fact that the calculations are done only with a limited set of condensing substances, likely $H_2SO_4$. It is well known that the exclusive condensation of $H_2SO_4$ is too slow for the growth rates observed in the atmosphere and that other substances like VOC's or ELVOC's have a large share on growth rates (Ehn et al, 2014; Kuang et al, 2012; Kupc et al, 2020 and literature cited there). An overview about the problems associated with the investigation of nucleation under Chinese high pollution conditions is given by Chu et al (2019). Here also potential other compounds that might be important for NPF. Cai et al (2017) showed that $H_2SO_4$ could and should be involved but, high $H_2SO_4$ concentrations do not necessary lead to particle formation events.

The physical background besides the description of the mathematical steps is largely missing. Neither the environmental conditions for the validity of the model, nor the initial input parameters, temperatures humidity, number and size of large particles acting as a condensation sink etc. are included. 100 nm particles are mentioned in a 'certain' number.

The major difference between the new approach of the 'appearance time method' and the 'old' theory, that needs to be discussed in detail is the result that, for the small particles size the growth rate due to coagulation is slowed down compared to other studies where the growth rate is continuously increasing with size (Kulmala et al, 2004, Kulmala et al, 2013, Kuang et al, 2012). The result is a longer appearance time (12 hour for 4.9 nm) than published elsewhere (Kulmala et al, 2013). For a detailed description of the current understanding of the initial steps of gas to particle conversion, and probably the 'theory' mentioned in the manuscript see also Kulmala et al, 2017 and Yao et al, 2018.

The model results are then compared to an event 'typical' for Beijing, with nucleation mode particles observed in late winter (24.2.2018). Also, in this section the authors do not give any details about the environmental conditions at the day of the particle event, nor any detailed information about the location, distance to major roads, height of the aerosol inlet above ground etc.. This information is important in a study about aerosols that may be affected by strong local sources (traffic, home emissions) and potential regional transport. Included is only a reference to a submitted, but still not yet accepted, companion paper. Also, as this is an aerosol study under conditions with a large condensational sink, it would be helpful to have data on fine particle concentration, a 'certain number of 100 nm particles' is way too uncertain.

The most important issue, however, is the classification of the experimental comparison data. It is the timing of the event that makes it difficult to match the event with basic atmospheric physics and chemistry and with the current observations that the mayority of NPF events are happening during daylight hours. A new particle formation event at that time of the day would require an even faster growth than shown by Kulmala (2013) and not a slower growth as indicated in this manuscript. Both, the 'old' Kulmala 4 hours and the 'new' 12 hours appearance time require nighttime chemistry and physics which has not been reported as a significant source for nucleation.

The data presented rather indicate a transport related change of air masses. The
H2SO4 concentration which is at background levels at sunrise peaks about one hour after sunrise and than steadily declines throughout the day (Fig. 6 of the manuscript). This behavior is contrary to the model where the peak concentration is 4 hours later at noon. This experimental diurnal pattern is neither in agreement with local emissions in a high NOx environment nor with a photochemical reaction including OH radicals and SO2.

Following Kulmala et al (2013) the appearance of 5 nm particles needs about 4 hours after initial cluster formation. This implies for the Beijing case study that cluster formation would occur before sunrise. The air mass at that time has been, according to HYSPLIT, in the area close to the city of Tangshan, not locally in Beijing. Following the results of the current study (appearance time of 4.9 nm particles ∼12 hours) clusters would originate even further to the east, close to the coastline at a formation time a few hours after sunset (Fig. 1). This requires a discussion of a nighttime cluster formation chemistry.

Along the backtrajectories several anthropogenic sources for sulphur compounds are located, that emit a mixture of both, particles and gas phase precursors (Fig. 1). Their emissions include sulfur dioxide, H2SO4 (Srivastava et al, 2004) and large amounts of ammonia (Li et al, 2017) and primary nanoparticles produced at elevated temperatures and with or without catalysts, Bai et al, 1992). These are than released into elevated layers of the atmosphere (Junkermann et al, 2011). Measured emission rates of such 'new aerosol generators' are in the order of ∼3*10ˆ15 s-1 MW-1. Naturally, the emitted particles undergo coagulation and further growth or shrinking (evaporation) depending on the ambient conditions (temperature, humidity) during transport under cool and dry nighttime conditions and the plume conditions favor additional gas to particle conversion (Mohnen and Lodge, 1969). The shape of the diurnal pattern of H2SO4 in Beijing is in good agreement with such an advection-convection driven transport, by far better than with a local chemistry initiated cluster formation.

The example of the early morning particle event in Beijing is probably a typical

'nanoparticle' but not a formation event. The experimental verification is looking reasonable but likely does not reflect reality.

The manuscript would need a complete revision due to the missing model and experiment description and not verified nor discussed assumptions about the classification of the particle event. It's also required to discuss the differences between the current model and the previously published growth rate versus particle sizes and the implications on the timing of the whole nucleation process (cluster formation at night versus photochemical reaction).

The authors also should make sure that the appearance of nucleation mode particles in Beijing is applied to a really local phenomena and not to a plume study.

Specific comments:

Model output

The model based on the new formula predicts initially a slow down of the growth rates of new particles within the size range from 1 to 3 nm.

The behavior of the growth rates in the 'new' model setup of the manuscript would delay the growth of newly formed particles significantly compared growth rates published elsewhere (Kulmala et al, 2004 and Kulmala et al, 2013). In these publications a continuously increasing growth rate is used. Accordingly, with the new model the growth of new particles to sizes of about 5 nm takes $\sim$ 10-12 hours compared to 4 hours in the Kulmala et al (2013) example. This is a severe difference likely due to the restriction of only one condensing vapor and has to be discussed in detail because it has strong implications for the time and location of the cluster formation process.

This difference implies several questions that are important for the model - experiment comparison (probably not for the calculation of the impact of coagulation but for the selection of a nucleation event and nocturnal chemistry) that arise, mainly related to timing:

- What is the process producing the initial clusters? At what environmental conditions temperature, rH, radiation?

- Where (locally) does cluster formation happen and, in case, $H_2SO_4$ is involved, where is it's origin?

- Are other compounds required, ammonia, water etc..?

- Why is cluster formation happening at this time and the corresponding chemical reactions are terminated after a few hours while precursors are still present?

- The model is running at least for one version with constant vapor concentration. Is this real?

- When $H_2SO_4$ is crucial for the cluster formation, why are there no more 5 nm particles appearing 12 hours after the peak of $H_2SO_4$?

- Why is the assumed $H_2SO_4$ concentration as high as measured peak values? Within the hours when nucleation mode particles grow in the model from 1 to 3 nm the $H_2SO_4$ concentration is only a smaller fraction of the model value. $H_2SO_4$ reaches model values at a time when already 10 nm particles are detected.

3D transport:

Looking into the experimental data given in Fig. 7 transport of externally produced particles and precursors is more likely to explain the observations (Cai et al 2018). Transport at night from elevated sources ($\sim$ 300 m) normally happens in clear air (low condensation sink) above the polluted nocturnal surface layer (Fig. 2). Vertical convection mixes rapidly (< 30 min) the air of residual and surface layers in the morning (Platis et al, 2016; Junkermann and Hacker, 2018). Contrary to atmospheric processes industrial nanoparticle sources run 24 hours, seven days a week and include sulphur and nitrogen chemistry as well as ammonia in large amounts (Bai et al, 1992).

A simple budget calculation with a box model similar like in Cai et al (2018) for a 300 km

plume originating at Suizhong and Quinhuangdao spreading to ∼40 km width (Rosenfeld et al, 2000), assuming a few hundred m residual layer (Platis et al, 2016, Junkermann and Hacker, 2018) and a replacement of the plume volume above Beijing with a wind speed of 25 km h-1 results, even without additional new particles, in number concentrations already in the order of magnitude finally observed in Beijing (> 80000 cm-3) and in H2SO4 concentrations ∼ 10ˆ8 cm-3, see also Junkermann and Hacker (2018). It is also in agreement with the diurnal pattern (Fig. 6) with convection beginning at ∼08:00 local time and downwards mixing of a 200 m layer of this air mass into the low nucleation mode particle concentration 600 m surface layer (HYSPLIT) (Fig. 3). Numbers are based on measured and published emission data for power stations and from HYSPLIT meteorological data along the trajectory and allow at least a rough estimate of the magnitude of concentrations. How far new particle formation plays a role during the dry, cold and dark conditions during the transport and how aerosol size distributions and H2SO4 concentrations change would be a perfect task for a complete aerosol-chemistry-transport model. Such a plume production (Mohnen and Lodge, 1969) would widen the nanoparticle size distribution and provide additional particles in the lowest size bins. Field data show, that nighttime transport in the residual layer does not really lead to a massive loss of particles (Fig. 4). The plume hypothesis above is also in agreement with the results of spatial measurements of nucleation mode particles in Germany (Ma and Birmili, 2015) at stations surrounded by several fossil fuel burning power stations using SCR or SNCR technology.

Minor comments

Who is Julia (Page 7, line 1)

A reference to a paper under review (Cai et al, 2020) still has to be considered as grey literature as long as the review process is not open access. Also, as it is not clear whether the companion manuscript passes the review process, the information in this other paper is crucial for the current manuscript but might not be available. It could be placed in a supplement.

[Figure]

A reference to a paper in preparation (Li and Cai, 2020) is not even grey literature!

Define the conventional method, when it is first cited, Page 5, Lethipalo, 2014? Later in the paper it is confusing, with which model the current results are compared without a reference.

The same holds for the theory when the model, which is theory as well, is compared to another theory. Within the time has to be defined in the text and figures. Is this Beijing CST?

Numbering of the tables. Why is the third table named table A1 instead of Table 3?

References:

Bai, H., Biswas, P., and Keener, T.: Particle Formation by NH3-SO2 Reactions at trace water conditions, Ind. Eng. Chem. Res., 31, 88–94, https://doi.org/10.1021/ie00001a013, 1992.

Cai, R., Yang, D., Fu, Y., Wang, X., Li, X., Ma, Y., Hao, J., Zheng, J., and Jiang, J.: Aerosol surface area concentration: a governing factor in new particle formation in Beijing, Atmos. Chem. Phys., 17, 12327–12340, https://doi.org/10.5194/acp-17-12327-2017, 2017

Cai, R., Chandra, I., Yang, D., Yao, L., Fu, Y., Li, X., Lu, Y., Luo, L., Hao, J., Ma, Y., Wang, L., Zheng, J., Seto, T., and Jiang, J.: Estimating the influence of transport on aerosol size distributions during new particle formation events, Atmos. Chem. Phys., 18, 16587–16599, https://doi.org/10.5194/acp-18-16587-2018, 2018

Chu, B., Kerminen, V.-M., Bianchi, F., Yan, C., Petäjä, T., and Kulmala, M.: Atmospheric new particle formation in China, Atmos. Chem. Phys., 19, 115–138, https://doi.org/10.5194/acp-19-115-2019, 2019

Dunne, E. M., Gordon, H., Kürten, A., Almeida, J., Duplissy, J., Williamson, C., Ortega, I. K., Pringle, K. J., Adamov, A., Baltensperger, U., Barmet, P., Benduhn, F., Bianchi, F.,

Breitenlechner, M., Clarke, A., Curtius, J., Dommen, J., Donahue, N. M., Ehrhart, S., Flagan, R. C., Franchin, A., Guida, R., Hakala, J., Hansel, A., Heinritzi, M., Jokinen, T., Kangasluoma, J., Kirkby, J., Kulmala, M., Kupc, A., Lawler, M. J., Lehtipalo, K., Makhmutov, V., Mann, G., Mathot, S., Merikanto, J., Miettinen, P., Nenes, A., Onnela, A., Rap, A., Reddington, C. L. S., Riccobono, F., Richards, N. A. D., Rissanen, M. P., Rondo, L., Sarnela, N., Schobesberger, S., Sengupta, K., Simon, M., Sipilä, M., Smith, J. N., Stozkhov, Y., Tomé, A., Tröstl, J., Wagner, P. E., Wimmer, D., Winkler, P. M., Worsnop, D. R., and Carslaw, K. S.: Global atmospheric particle formation from CERN CLOUD measurements, Science, 354, 1119- 1124, 10.1126/science.aaf2649, 2016.

Ehn, M., Thornton, J. A., Kleist, E., Sipila, M., Junninen, H., Pullinen, I., Springer, M., Rubach, F., Tillmann, R., Lee, B., Lopez-Hilfiker, F., Andres, S., Acir, I.-H. Rissanen, M., Jokinen, T. Schobesberger, S., Kangasluoma, J., Kontkanen, J., Nieminen, T., Kurten, T., Nielsen, L. B., Jørgensen, S., Kjaergaard, H. G., Canagaratna, M., Dal Maso, M., Berndt, T., Petäjä, T., Wahner, A., Kerminen, V-M., Kulmala, M., Worsnop, D. R., Wildt, J., and Mentel, T. F.: A large source of low-volatility secondary organic aerosol, Nature, 506, 476–479, https://doi.org/10.1038/nature13032, 2014.

Junkermann, W., Vogel, B. and Sutton, M.A., The climate penalty for clean fossil fuel combustion, Atmos. Chem. Phys, 11, 12917-12924, 2011

Junkermann W., and Hacker J., 2018, Ultrafine particles in the lower troposphere: major sources, invisible plumes and meteorological transport processes, BAMS, 99, 2587-2622, Dec. 2018, DOI:10.1175/BAMS-D-18-0075.1

Kulmala, M., Laakso, L., Lehtinen, K. E. J., Riipinen, I., Dal Maso, M., Anttila, T., Kerminen, V.-M., Hõrrak, U., Vana, M., and Tammet, H.: Initial steps of aerosol growth, Atmos. Chem. Phys., 4, 2553–2560, https://doi.org/10.5194/acp-4-2553-2004, 2004.

Kulmala, M., Lehtinen, K. E. J., and Laaksonen, A.: Cluster activation theory as an explanation of the linear dependence between formation rate of 3 nm particles and sulphuric acid concentration, Atmos. Chem. Phys., 6, 787–793,

https://doi.org/10.5194/acp-6-787-2006, 2006.

Kulmala, M., Kontkanen, J., Junninen, H., Lehtipalo, K., Manninen, H. E., Nieminen, T., Petäjä, T., Sipilä, M., Schobesberger,S., Rantala, P., Franchin, A., Jokinen, T., Järvinen, E., Äijälä, M., Kangasluoma, J., Hakala, J., Aalto, P. P., Paasonen, P., Mikkilä, J., Vanhanen, J., Aalto, J., Hakola, H., Makkonen, U., Ruuskanen, T., Mauldin III., R. L., Duplissy, J., Vehkamäki, H., Bäck, J., Kortelainen, A., Riipinen, I., Kurten, T., Johnston, M. V., Smith, J. N., Ehn, M., Mentel, T., Lehtinen, K. E. J., Laaksonen, A., Kerminen, V.-M., and Worsnop, D. R.: Direct observations of atmospheric aerosol nucleation, Science, 339, 943–946, https://doi.org/10.1126/science.1227385, 2013

Kulmala, M., V.-M. Kerminen, T. Petäjä, A. J. Ding, L. Wang. Atmospheric gas-to-particle conversion: why NPF events are observed in megacities? Faraday Discussions, 2017; 200: 271 DOI: 10.1039/c6fd00257a Kupc, A., Williamson, C. J., Hodshire, A. L., Kazil, J., Ray, E., Bui, T. P., Dollner, M., Froyd, K. D., McKain, K., Rollins, A., Schill, G. P., Thames, A., Weinzierl, B. B., Pierce, J. R., and Brock, C. A.: The potential role of organics in new particle formation and initial growth in the remote tropical upper troposphere, Atmos. Chem. Phys. Discuss., https://doi.org/10.5194/acp-2020-675, in review, 2020.

Lehtipalo, K., Leppä, J., Kontkanen, J., Kangasluoma, J., Franchin, A., Wimmer, D., Schobesberger, S., Junninen, H., Petäjä, T., Sipilä, M., Mikkilä, J., Vanhanen, J., Worsnop, D. R. & Kulmala, M. 2014: Methods for determining particle size distribution and growth rates between 1 and 3 nm using the Particle Size Magnifier. Boreal Env. Res. 19 (suppl. B): 215–236.

Li, Z., Jingkun Jiang, Zizhen Ma, Oscar A. Fajardo, Jianguo Deng, Lei Duan, Influence of flue gas desulfurization (FGD) installations on emission characteristics of PM2.5 from coal-fired power plants equipped with selective catalytic reduction (SCR), Environmental Pollution, Volume 230, 2017, 655-662, ISSN 0269-7491, https://doi.org/10.1016/j.envpol.2017.06.103.

[Figure]

Ma, N., and Birmili, W., 2015 Estimating the contribution of photochemical particle formation to ultrafine particle number averages in an urban atmosphere, Science of the total environment, Vol 521-513; 154-166

Mohnen V.A. and Lodge, J.P., General survey of gas-to-particle conversions, Proc. 7th ICCN, Prague (1969)

Napari, I., Noppel, M., Vehkamäki, H., and Kulmala, M.: Parametrization of ternary nucleation rates for H2SO4-NH3-H2O vapors, J. Geophys. Res., 107, AAC 6-1-AAC 6-6, 10.1029/2002jd002132, 2002

Platis, A., and Coauthors, 2015: An Observational Case Study on the Influence of Atmospheric Boundary-Layer Dynamics on New Particle Formation, Boundary-Layer Meteorology, DOI 10.1007/s10546-015-0084-y

Riccobono, F., Schobesberger, S., Scott, C. E., Dommen, J., Ortega, I. K., Rondo, L., Almeida, J., Amorim, A., Bianchi, F., Breitenlechner, M., David, A., Downard, A., Dunne, E. M., Duplissy, J., Ehrhart, S., Flagan, R. C., Franchin, A., Hansel, A., Junninen, H., Kajos, M., Keskinen, H., Kupc, A., Kürten, A., Kvashin, A. N., Laaksonen, A., Lehtipalo, K., Makhmutov, V., Mathot, S., Nieminen, T., Onnela, A., Petäjä, T., Praplan, A. P., Santos, F. D., Schallhart, S., Seinfeld, J. H., Sipilä, M., Spracklen, D. V., Stozhkov, Y., Stratmann, F., Tomé, A., Tsagkogeorgas, G., Vaattovaara, P., Viisanen, Y., Vrtala, A., Wagner, P. E., Weingartner, E., Wex, H., Wimmer, D., Carslaw, K. S., Curtius, J., Donahue, N. M., Kirkby, J., Kulmala, M., Worsnop, D. R., and Baltensperger, U.: Oxidation products of biogenic emissions contribute to nucleation of atmospheric particles, Science, 344, 717-721, 10.1126/science.1243527, 2014.

Rosenfeld D., 2000: Suppression of Rain and Snow by Urban and Industrial Air Pollution, Science, 287, 1793 Yao, L., Olga Garmash, Federico Bianchi, Jun Zheng, Chao Yan, Jenni Kontkanen, Heikki Junninen, Stephany Buenrostro Mazon, Mikael Ehn, Pauli Paasonen, Mikko Sipilä, Mingyi Wang, Xinke Wang, Shan Xiao, Hangfei Chen, Yiqun Lu, Bowen Zhang, Dongfang Wang, Qingyan Fu, Fuhai Geng, Li Li, Hongli

Wang, Liping Qiao, Xin Yang, Jianmin Chen, Veli-Matti Kerminen, Tuukka Petäjä, Douglas R. Worsnop, Markku Kulmala, Lin Wang. Atmospheric new particle formation from sulfuric acid and amines in a Chinese megacity. Science, 2018; 361 (6399): 278 DOI: 10.1126/science.aao4839

Tian, H., Kaiyun Liu, Jiming Hao, Yan Wang, Jiajia Gao, Peipei Qiu, Chuanyong Zhu Nitrogen Oxides Emissions From Thermal Power Plants in China: Current Status and Future Predictions PMID: 24010996 DOI: 10.1021/es402202d

Zhang, Y. Wang, T., Pan, W.P., Romero C.E., Advances in Ultra-low Emission Control Technologies for Coal-Fired Power Plants, Woodhead publishing, Elsevier, 2019, ISBN 9780-0-08-102419-5

[Figure]

[Figure]

**Fig. 1.** Fig. 1: HYSPLIT 12 h backtrajectories for 24022018 01:00 UTC arrival in Beijing, yellow 50 m, red 250 m and green 500 m, GDAS 0.5

[Figure]

**Fig. 2.** Fig. 2: Febr. 19, 2002 early morning (09:00) conditions 500 m above the Po-Valley (Italy)

[Figure]

**Fig. 3.** Fig. 3: Boundary layer development in Beijing the morning of Febr. 24, 2018 (HYSPLIT)

[Figure]

**Fig. 4.** Fig. 4:Size distribution above and below inversion 100 km, 10 h downwind of source (Germany, June 10, 2014, PBL 600 m), clean air in the residual layer, polluted in the PBL

---

## Referee Comment (RC2) · Anonymous Referee #2 · 11 Aug 2020

General Comments:

This manuscript investigates the impact of coagulation on the particle growth rate, calculated using the appearance time method, using theoretical derivations and aerosol dynamics modeling. The topic of actual growth rate calculation is of great importance for understanding the new particle formation processes in the atmosphere. The appearance time method was originally developed by Lehtipalo et al., 2014 to calculate growth rate in the size range 1-3nm using PSM data. They highlighted that the method is robust unless coagulation process affect greatly the particles size distributions, such as a heavily polluted environment with high number concentration of preexisting parti-

cles. Although this paper is presenting a correction for coagulation on the appearance time method, the approach and the validity of the method are not adequately described. It would be more appropriate to first describe the method and its weaknesses, then present the suggested correction for coagulation impacts and then apply the method in different environment types (boreal forest and Beijing data are available for comparison). These steps, are only briefly described and definitely more examples need to be presented.

Specific comments:

Page 2, Line 19: Only few or just one application?

Page 4, Line 13 and 25 and further: Coagulation coefficient unit should be cm3 s-1.

Page 5, Line 24: As this paper describes a correction to the appearance time method it is proper to present the method.

Page 6, Line 17: This section is lacking all the necessary information for the reader to understand the theoretical and experimental tools that were used to perform this study.

Page 7, Line 1: What is Julia?

Page 7, Line 20: This reference is a paper under review. More details about the experimental part should be given here.

Page 7, Line 23: This section (4.1) needs to be moved to methods, where the appearance method should be described and cited.

Page 11, Line 14: This is not shown here, we have no indication about sub-1.3nm growth rates.

Page 11, Line 15:In Figure 4, Coags corrected seems to perform better than the Corrected total growth for particles larger than 3nm.

Page 11, Line 26: This paragraph is confusing. Which formula is used, Eq 5, or some

other formula from the Appendix?

Page 12, Line 4: It has to be shown here that the method is described so that all limitations are discussed prior to applying the new method.

Page 12, Line 13: It is the previous study, or are there more studies?

Page 12, Line 24: I would not use the expression agrees better, as it does not seem to agree, it seems to work better than the conventional method but still overestimates all particle growth rates outside the range 2-3.5 nm. These discrepancies both in absolute values but also with regard to increasing size and especially the shape of the curve have to be discussed further in the text. It has to be noted that the new curve has the same shape as the uncorrected one which suggests that there is an underlying assumption causing these deviations, it is worth providing more information.

Page 12, Line 34: This assumption is valid for cases with clear diurnal variations of vapor concentrations as the assumed one. However what happens when condensing species exist in the afternoon as well, then the GR would be much higher. The example in 4.4 is demonstrating this weakness as in the afternoon the GR calculation is three time higher. Sensitivity tests with condensing species not vanishing in the afternoon could be useful as well.

Page 13, Line 4: A single NPF event is not enough to demonstrate the validity of the proposed correction. Different events, under various meteorological and environmental conditions and under different environment types (and hence condensing species) are necessary to my opinion to test the new formulae.
* * *

---

## Author Comment (AC1) · 11 Sep 2020

**Responses to Reviewer #1's Comments on Manuscript acp-2020-398**

(Impacts of coagulation on the appearance time method for sub-3nm particle growth rate evaluation and their corrections)

We thank Prof. Dr. Wolfgang Junkermann (referred as reviewer #1 below) for the efforts and constructive comments that help to improve this manuscript. The reviewer's comments are addressed in the following paragraphs and the manuscript were revised majorly. In response to the concerns of reviewer #1, we clarify in the revised section 3.1 that the numerical models in this study are used to provide a benchmark to test the proposed formulae rather than investigate the nucleation mechanism in the real atmosphere. The comments are shown as sans-serif dark red texts and our responses are shown as serif black texts. Changes are highlighted in the revised manuscript and shown as "quoted underlined texts" in the responses. Line numbers, figures, and equations quoted in the responses correspond the revised manuscript. References are given at the end of the responses.

**Reviewer #1**

Summary

The manuscript describes a numerical set of formulas that is used in an aerosol model for the calculation of the growth of aerosol particles after nucleation, especially for the range of ~ 1 nm to 3 nm. This formula set is used for a calculation of the appearance time (time after cluster formation) and later for estimation of the growth rate of the larger sizes aerosols > 4 nm although the main emphasis is on the smallest size ranges, where coagulation has a major effect. Model results based on these formulas are finally compared to a nucleation mode particle event in a relative highly polluted environment in China, in the city of Beijing in a late winter situation.

**Response**:

This manuscript describes the formulae to correct the influence of coagulation on the appearance time method for growth rate estimation, especially for the range of ~1-3 nm. The derivations of the conventional and corrected appearance time methods for growth rate estimation are detailed in Sections 4.1 and 4.2. In addition to derivations, we also test these formulae for the conventional and corrected appearance time methods using aerosol dynamic models (Section 4.1-4.3, Figs. 1-6). The comparison between the growth rate given by the model and the rates retrieved for the simulated evolution of aerosol size distribution using the conventional and appearance time method support the argument that the impact of coagulation should be corrected when using the appearance time method to estimate the new particle growth rate. Further, we provide an example to illustrate the effect of coagulation on the growth rate retrieved using the appearance time method in the real atmosphere in urban Beijing. A comparison between the growth rates retrieved using the conventional and corrected appearance time method is shown during a new particle formation event in Beijing. There is no comparison between the measured size distributions in urban Beijing and the simulated distributions using the aerosol dynamics models in this study. These research methods are also summarized in the last paragraph of the Introduction section.

According to the reviewer's comments below, a majority of the reviewer's concerns are closely related to the comment "Model results based on these formulas are finally compared to a nucleation mode particle event". Detailed responses will be given following every comment and we present a summary of these responses in the following paragraph.

In the revised manuscript, we clarified that models used in this study are based on aerosol dynamics but not the formulae for conventional or corrected appearance time methods. A paragraph is added at the beginning of Section 3 for

a better understanding. The input of the models are an initial aerosol size distribution and the concentration of the gaseous precursor as a function of time. The evolution of the aerosol size distribution is obtained by solving the aerosol general dynamic equations. Meanwhile, particle growth rate as a function of particle size and time is given by the aerosol dynamic model, which is calculated according to the condensation, evaporation, and coagulation rates. The formulae for conventional or corrected appearance time methods are used to retrieve particle growth rate from the simulated evolution of the aerosol size distribution. Since the simulated growth rate (given by the aerosol dynamic model) is consistent with the simulated evolution of the aerosol size distribution while there are approximations in the estimation using the appearance time methods, the comparison between the simulated growth rate and the growth rate retrieved using the appearance time methods indicates the uncertainty of the appearance time methods in the simulation conditions. The simulation conditions are chosen to elaborate not only the accurateness and uncertainties of the appearance time methods but also the causes of these uncertainties. For instance, Fig. 5 shown the accuracy of the corrected appearance time method under a constant vapor concentration, which support the derivations for the appearance time method. However, since the derivations are based on a constant vapor concentration assumption whereas the vapor concentration in real new particle formation events may vary with time, we present Fig. 6 and Table A1 to shown the uncertainties due to this violation of the constant vapor assumption. After testing the appearance time methods using the aerosol dynamics models, the appearance time methods are applied in a new particle formation event measured in urban Beijing. The growth rate during this event are estimated using the conventional and corrected appearance time methods and their differences indicate the influences of the coagulation on the appearance time of new particles. The measured size distributions and the retrieved growth rates for this new particle formation event are not compared to those given by the aerosol dynamic models.

We revised the first sentence in Section 3.1 as "A discrete aerosol model and a discrete-sectional aerosol model based on aerosol dynamics were used to provide an evolving aerosol size distribution and hence to test the conventional and corrected appearance time methods" and added "Note that the aerosol dynamic models are only used to provide a benchmark to compare the conventional and corrected appearance time methods in this study" at the end of this section.

**General comments**

The argumentation and description of the formulas seems plausible and is according to the authors in agreement with the theory. Unfortunately this 'reference' theory, which is several times cited in the manuscript, is neither described nor referenced. Also there is no information about this general 'theory' and the underlying nucleation scheme applied. There are several nucleation schemes, binary nucleation of sulfuric acid and water (neutral and ion-assisted), ternary involving sulfuric acid, water, and ammonia; with or without organic species and sulfuric acid or charged sulfuric acid-water-ammonia (Napari et al, 2002; Riccobono et al. 2014; Dunne et al. 2016). Otherwise, the results of the subsequent calculations are at least partially contradicting previous publications that are a base for many 'new particle formation' studies within the last decade.

That discrepancy is likely due to the fact that the calculations are done only with a limited set of condensing substances, likely $H_2SO_4$. It is well known that the exclusive condensation of $H_2SO_4$ is too slow for the growth rates observed in the atmosphere and that other substances like VOC's or ELVOC's have a large share on growth rates (Ehn et al, 2014; Kuang et al, 2012; Kupc et al, 2020 and literature cited there). An overview about the problems associated with the investigation of nucleation under Chinese high pollution conditions is given by Chu et al (2019). Here also potential other compounds that might be important for NPF. Cai et al (2017) showed that $H_2SO_4$ could and should be involved but, high $H_2SO_4$ concentrations do not necessary lead to particle formation events.

**Response**: We agree with the reviewer that the nucleation and condensation of a single vapor (e.g., $H_2SO_4$) is usually insufficient to explain new particle formation in the atmosphere. However, the aim of this study is to investigate how to

estimate the growth rate from the measured aerosol size distributions rather than figure out the compounds contribute to particle growth and the their growth mechanisms. The aerosol dynamic models used to test the formulae of the appearance time methods and they are not used to illustrate the nucleation and growth mechanism. Hence, the simplifications in the aerosol dynamics models in terms of the nucleation mechanism does not affect the tests because they do not affect the consistency between the growth rate and evolution of the aerosol size distribution given by the model. There is no modeling results in the application example in Fig. 7.

A new paragraph was added at the beginning of Section 3 to clarify the reason to use models.

Although it does not influence the statement of the reviewer in this piece of comment, we would like to briefly clarify the conclusion of a previous study (Cai et al., 2017). The reason that high $H_2SO_4$ concentrations do not necessary lead to particle formation events in urban Beijing is usually because of the aerosol Fuchs surface area concentration (coagulation sink of new particles). When $H_2SO_4$ concentration was high and coagulation sink was low, there were usually new particles observed in urban Beijing. However, this does not indicate that $H_2SO_4$ is the only major compound participate new particle formation in urban Beijing and the contributions of other compounds to particle growth were discussion in Cai et al. (2017).

The physical background besides the description of the mathematical steps is largely missing. Neither the environmental conditions for the validity of the model, nor the initial input parameters, temperatures humidity, number and size of large particles acting as a condensation sink etc. are included. 100 nm particles are mentioned in a 'certain' number.

**Response**: To clarify the background particles used in the aerosol dynamic model, we added "The condensation sink ($10^{-3}$ $– 10^{-2}$ s$^{-1}$) is contributed simultaneously by a certain number concentration ($7.2\times10^2 – 7.2\times10^3$ cm$^{-3}$) of 100 nm background particles and the new particles." Other input parameters (e.g., temperature) are omitted because they does not affect the mathematical derivations and the aerosol dynamics models are only used to test these derivations in this study.

The major difference between the new approach of the 'appearance time method' and the 'old' theory, that needs to be discussed in detail is the result that, for the small particles size the growth rate due to coagulation is slowed down compared to other studies where the growth rate is continuously increasing with size (Kulmala et al, 2004, Kulmala et al, 2013, Kuang et al, 2012). The result is a longer appearance time (12 hour for 4.9 nm) than published elsewhere (Kulmala et al, 2013). For a detailed description of the current understanding of the initial steps of gas to particle conversion, and probably the 'theory' mentioned in the manuscript see also Kulmala et al, 2017 and Yao et al, 2018.

**Response**: We agree with the reviewer that correcting the effect of coagulation reduces the value of growth rate retrieved using the appearance time method. However, there is no discrepancy of the size dependency of growth rate reported in this study and in previous studies. Figure 7b shows that the growth rate estimated using the corrected appearance time method generally increase with the particle size, while the slightly negative size dependency in the sub-3 nm size range is mainly due to the decreasing $H_2SO_4$ concentration and the size dependency of the coagulation coefficient.

The apparent relationship between particle growth rate and particle size does not necessarily characterized the size dependency of particle growth rate. In the revised section 2.3, we clarified that "Note that since the appearance time is a function of $d_p$ and $GR_{conv}$ is estimated for difference appearance time, the apparent relationship between $GR_{conv}$ and $d_p$ does not necessarily indicate the size dependency of $GR_{conv}$ at any given moment." For example, the methods in Kulmala et al. (2004, 2013) and this study only provides the growth rate as a function of both particle diameter and its corresponding appearance time. One has to exclude the influence of time-dependent variable (e.g., the varying vapor

concentration) to obtain the size dependency of growth rate. In contrast, Kuang et al. (2012) estimates the size dependency of particle growth rate using a time-and-size-resolved method via solving the aerosol general dynamic equation.

The "longer appearance time (12 hour for 4.9 nm)" refers to Fig. 6, in which the model was used to test the formulae. Hence, the simulated aerosol size distributions in this figure should not be directly compared to the measured size distributions in an atmospheric new particle formation event. Further, the appearance time corresponding to a particle size, by its definition, is a moment rather than a duration. We revised "12 h" as "12:00" to avoid confusion. As shown in Fig. 6, the duration for a particle to grow from 1.0 nm to 4.9 nm is ~2.5 h, which is comparable to the values in previous studies. However, this value is mainly determined by the input $H_2SO_4$ concentration and it has little contribution to understanding the nucleation mechanism because the models in this study are only used to test the formulae.

The model results are then compared to an event 'typical' for Beijing, with nucleation mode particles observed in late winter (24.2.2018). Also, in this section the authors do not give any details about the environmental conditions at the day of the particle event, nor any detailed information about the location, distance to major roads, height of the aerosol inlet above ground etc.. This information is important in a study about aerosols that may be affected by strong local sources (traffic, home emissions) and potential regional transport. Included is only a reference to a submitted, but still not yet accepted, companion paper. Also, as this is an aerosol study under conditions with a large condensational sink, it would be helpful to have data on fine particle concentration, a 'certain number of 100 nm particles' is way too uncertain.

**Response**: The example of the new particle formation event in urban Beijing is used to elaborate that under the relatively high coagulation sink in urban Beijing, the influence of coagulation on the appearance time of new particles and hence the retrieved growth rate is non-negligible for sub-3 nm particles. The growth rate estimated using the appearance time method in Fig. 7b was calculated using the measured aerosol size distributions. The measured size distributions in Fig. 7a were not compared to the simulated distributions in Figs. 1-6.

To clarify the environment conditions during this new particle formation event, we added "During 8:00 – 16:00 on this NPF day, the mean temperature, relative humidity, and wind speed are 1.3 °C, 22%, and 1.4 m s$^{-1}$, respectively" in section 3.2. A recently published manuscript, Deng et al. (2020b), reported the data measured at the same site during the same period. This reference was included to provide detailed information on the measurement site.

The measured aerosol size distribution ranging from 1-400 nm are shown in Fig. 7a. Note that "a certain number of 100 nm" particle were only used in the simulation shown in Figs. 3b – 6, and the simulation was not compared to measurements.

The most important issue, however, is the classification of the experimental comparison data. It is the timing of the event that makes it difficult to match the event with basic atmospheric physics and chemistry and with the current observations that the majority of NPF events are happening during daylight hours. A new particle formation event at that time of the day would require an even faster growth than shown by Kulmala (2013) and not a slower growth as indicated in this manuscript. Both, the 'old' Kulmala 4 hours and the 'new' 12 hours appearance time require nighttime chemistry and physics which has not been reported as a significant source for nucleation.

**Response**: We have revised "12 h" as "12:00". As shown in Fig. 7a, it took ~2 h for particles at 1.5 nm to grow up to 5 nm during the event on Feb. 24, 2018.

The data presented rather indicate a transport related change of air masses. The $H_2SO_4$ concentration which is at background levels at sunrise peaks about one hour after sunrise and then steadily declines throughout the day (Fig. 6 of the manuscript). This behavior is contrary to the model where the peak concentration is 4 hours later at noon. This experimental diurnal pattern is neither in agreement with local emissions in a high $NO_x$ environment nor with a photochemical reaction including OH radicals and $SO_2$.

**Response**: The $H_2SO_4$ concentration used in the model (Fig. 6) was an input parameter because the model was used only the test the appearance time method rather than predict the $H_2SO_4$ concentration. It was determined to be either constant or follow a normal distribution to investigate the influence of a varying vapor concentration on the appearance time method, as detailed in section 4.3.

We agree with the reviewer that transport, mixing, and local emissions may influence the measured $H_2SO_4$ concentration. However, due to the high condensation sink and hence short atmospheric residence time of $H_2SO_4$ in the presence of a relatively high aerosol concentration, the measured $H_2SO_4$ is more likely to form at the observation site than being transported from elsewhere. Figure R1 shows the diurnal variation of $SO_2$ concentration on Feb. 24, 2018. The steadily decreasing $H_2SO_4$ concentration in Fig. 7 is in accordance with the decreasing $SO_2$ concentration between 8:00-16:00, indicating that the photochemical reaction between OH radicals and $SO_2$ is a major pathway for the formation of $H_2SO_4$ during this new particle formation event.

[Figure]

Figure R1. $SO_2$ concentration measured at three nearest national monitoring stations on Feb. 24, 2018.

Following Kulmala et al (2013) the appearance of 5 nm particles needs about 4 hours after initial cluster formation. This implies for the Beijing case study that cluster formation would occur before sunrise. The air mass at that time has been, according to HYSPLIT, in the area close to the city of Tangshan, not locally in Beijing. Following the results of the current study (appearance time of 4.9 nm particles ~12 hours) clusters would originate even further to the east, close to the coastline at a formation time a few hours after sunset (Fig. 1). This requires a discussion of a nighttime cluster formation chemistry.

**Response**: As clarified above, Fig. 7a shows that during this new particle formation event, the appearance time for 1.5 nm and 5 nm aerosols are approximately 8:00 and 10:00, respectively. As clarified above, the 12-h growth is a misinterpretation of this manuscript. This appearance time and the measured aerosol size distribution show that new particles down to the cluster size were observed at the site after sunrise and they grew steadily into large particles. Such

a complete formation and growth pattern of new particles indicates a typical regional new particle formation event (Kulmala et al., 2012), although transport, mixing, and local emissions may influence the measured aerosol size distributions.

Along the back trajectories several anthropogenic sources for sulphur compounds are located, that emit a mixture of both, particles and gas phase precursors (Fig. 1). Their emissions include sulfur dioxide, $H_2SO_4$ (Srivastava et al, 2004) and large amounts of ammonia (Li et al, 2017) and primary nanoparticles produced at elevated temperatures and with or without catalysts, Bai et al, 1992). These are than released into elevated layers of the atmosphere (Junkermann et al, 2011). Measured emission rates of such 'new aerosol generators' are in the order of ~$3*10^{15}$ s$^{-1}$ MW$^{-1}$. Naturally, the emitted particles undergo coagulation and further growth or shrinking (evaporation) depending on the ambient conditions (temperature, humidity) during transport under cool and dry nighttime conditions and the plume conditions favor additional gas to particle conversion (Mohnen and Lodge, 1969). The shape of the diurnal pattern of $H_2SO_4$ in Beijing is in good agreement with such an advection-convection driven transport, by far better than with a local chemistry initiated cluster formation.

The example of the early morning particle event in Beijing is probably a typical 'nanoparticle' but not a formation event. The experimental verification is looking reasonable but likely does not reflect reality.

**Response**: We added a paragraph in Section 4.4 on the classification of the observed new particle formation event:

"As shown in Fig. 7, the measured $H_2SO_4$ concentration started to increase around 7:00 and its 50% appearance time was ~7:30. Because $SO_2$ concentration declined between 8:00 and 14:00, the peak concentration of $H_2SO_4$ was observed at 8:00 rather than at noon. Shortly after the increase in $H_2SO_4$ concentration, new particles down to the cluster size (~1 nm in geometric diameter) was observed. The high concentration of sub-2 nm aerosol during 7:00 - 15:00 and the continuous growth pattern of new particles from 1.5 nm to 10 nm indicates that the observed new particle formation event was a typical regional event (Kulmala et al., 2012), though transport, mixing, and emissions might influence the observed aerosol size distributions."

We disagree with the reviewer's hypothesis that the measured $H_2SO_4$ and aerosol size distribution at the site were predominated by transport. Due to the short atmospheric residence time of $H_2SO_4$ and sub-3 nm aerosols (Deng et al, 2020b), it is unlikely that they had been transported from somewhere 150 km away before the detection.

The agreement between the trends of the $H_2SO_4$ concentration in Fig. 7a and the $SO_2$ concentration in Fig. R1 supports the hypothesis that the daytime $H_2SO_4$ during this event was mainly formed by photochemical reaction. The continuous formation and growth pattern of particles down to the cluster size (~1 nm in geometric diameter) shown in Fig. 7a indicates the observed new particle formation event was a typical regional event. Furthermore, the agreement between the 50% appearance time of $H_2SO_4$ (7:30) and 1.5 nm particles (8:00) in Fig. 7a indicates that the $H_2SO_4$ formed after sunrise initiates the observed new particle formation event.

The manuscript would need a complete revision due to the missing model and experiment description and not verified nor discussed assumptions about the classification of the particle event. It's also required to discuss the differences between the current model and the previously published growth rate versus particle sizes and the implications on the timing of the whole nucleation process (cluster formation at night versus photochemical reaction).

The authors also should make sure that the appearance of nucleation mode particles in Beijing is applied to a really local phenomena and not to a plume study.

**Response**: Here we present a summary of revisions in response to the major concerns in the general comments of the reviewer.

- We clarified that the models in this study were used to test the proposed formulae for the appearance time method. There is no comparison between the modeling results and measurements in this study. Besides, we clarified the input background aerosol concentration for the model in the revised manuscript.

- We added more details on the experiments.

- We added a paragraph on the classification of the new particle formation event shown in Fig.7. The continuous formation of aerosols down to the cluster size and their subsequent growth are in accordance with the features of a regional new particle formation event rather than those of a plume event.

- The size-dependency of particle growth rate in the observed new particle formation event is similar to those reported in previous studies.

- The appearance time of $H_2SO_4$ and new particles on Feb. 24, 2018 indicates that the observed intensive new particle formation was mainly driven by daytime photochemical reaction.

**Specific comments:**

Model output

The model based on the new formula predicts initially a slow down of the growth rates of new particles within the size range from 1 to 3 nm. The behavior of the growth rates in the 'new' model setup of the manuscript would delay the growth of newly formed particles significantly compared growth rates published elsewhere (Kulmala et al, 2004 and Kulmala et al, 2013). In these publications a continuously increasing growth rate is used. Accordingly, with the new model the growth of new particles to sizes of about 5 nm takes ~10-12 hours compared to 4 hours in the Kulmala et al (2013) example. This is a severe difference likely due to the restriction of only one condensing vapor and has to be discussed in detail because it has strong implications for the time and location of the cluster formation process.

This difference implies several questions that are important for the model – experiment comparison (probably not for the calculation of the impact of coagulation but for the selection of a nucleation event and nocturnal chemistry) that arise, mainly related to timing:

- What is the process producing the initial clusters? At what environmental conditions temperature, RH, radiation?
- Where (locally) does cluster formation happen and, in case, $H_2SO_4$ is involved, where is its origin?
- Are other compounds required, ammonia, water etc..?
- Why is cluster formation happening at this time and the corresponding chemical reactions are terminated after a few hours while precursors are still present?
- The model is running at least for one version with constant vapor concentration. Is this real?
- When $H_2SO_4$ is crucial for the cluster formation, why are there no more 5 nm particles appearing 12 hours after the peak of $H_2SO_4$?
- Why is the assumed $H_2SO_4$ concentration as high as measured peak values? Within the hours when nucleation mode particles grow in the model from 1 to 3 nm the $H_2SO_4$ concentration is only a smaller fraction of the model value. $H_2SO_4$ reaches model values at a time when already 10 nm particles are detected.

**Response**: As clarified above, the concerns of the reviewer on the aerosol dynamic models and their results are mainly based on misunderstandings of this manuscript. We have revised the manuscript to avoid such confusion.

There is no comparison between the aerosol size distributions and growth rates given by the aerosol dynamic models and obtained from the field campaign in urban Beijing. The models are used to test the formula and they are not based on the formula. Besides, the models are not used to investigate nucleation and growth mechanisms. A continuously increasing

growth rate was observed in this study, which is in accordance with previous literature. It took ~2 h for a 1.5 nm particle to grow up to 5 nm during the observed new particle formation event on Feb. 24, 2018, and this duration of 2 h was estimated from the measured aerosol size distribution rather than predicted by a model.

3D transport:

Looking into the experimental data given in Fig. 7 transport of externally produced particles and precursors is more likely to explain the observations (Cai et al 2018). Transport at night from elevated sources (~300 m) normally happens in clear air (low condensation sink) above the polluted nocturnal surface layer (Fig. 2). Vertical convection mixes rapidly (< 30 min) the air of residual and surface layers in the morning (Platis et al, 2016; Junkermann and Hacker, 2018). Contrary to atmospheric processes industrial nanoparticle sources run 24 hours, seven days a week and include sulphur and nitrogen chemistry as well as ammonia in large amounts (Bai et al, 1992).

A simple budget calculation with a box model similar like in Cai et al (2018) for a 300 km plume originating at Suizhong and Qinhuangdao spreading to ~40 km width (Rosenfeld et al, 2000), assuming a few hundred m residual layer (Platis et al, 2016, Junkermann and Hacker, 2018) and a replacement of the plume volume above Beijing with a wind speed of 25 km h$^{-1}$ results, even without additional new particles, in number concentrations already in the order of magnitude finally observed in Beijing (> 80000 cm$^{-3}$) and in $H_2SO_4$ concentrations ~10ˆ8 cm$^{-3}$, see also Junkermann and Hacker (2018). It is also in agreement with the diurnal pattern (Fig. 6) with convection beginning at ~08:00 local time and downwards mixing of a 200 m layer of this air mass into the low nucleation mode particle concentration 600 m surface layer (HYSPLIT) (Fig. 3). Numbers are based on measured and published emission data for power stations and from HYSPLIT meteorological data along the trajectory and allow at least a rough estimate of the magnitude of concentrations. How far new particle formation plays a role during the dry, cold and dark conditions during the transport and how aerosol size distributions and $H_2SO_4$ concentrations change would be a perfect task for a complete aerosol-chemistry-transport model. Such a plume production (Mohnen and Lodge, 1969) would widen the nanoparticle size distribution and provide additional particles in the lowest size bins. Field data show, that nighttime transport in the residual layer does not really lead to a massive loss of particles (Fig. 4). The plume hypothesis above is also in agreement with the results of spatial measurements of nucleation mode particles in Germany (Ma and Birmili, 2015) at stations surrounded by several fossil fuel burning power stations using SCR or SNCR technology.

**Response**: We agree with the reviewer that 3D transport may influence the measured aerosol size distribution and gas concentrations, especially for large (e.g., >5 nm) particles and trace gases with relatively long atmospheric residence time (e.g., $SO_2$). However, considering the atmospheric residence time of $H_2SO_4$ and sub-1.5 nm aerosols, the measured data shown in Fig. 7a in this manuscript is most likely a new particle formation event rather than a plume event. Deng et al. (2020b) reported that in urban Beijing, the atmospheric residence time of sub-1.5 nm particles are usually less than 5 min because of the relatively high coagulation sink. Note that there is no coal-fired power plant or cement plant inside the fifth ring road of Beijing. As a result, the observed $H_2SO_4$ and sub-1.5 nm particles were most likely to be formed near the station via daytime photochemical reactions rather than be transported from kilometers away.

In a perspective from long-term measurements in urban Beijing, the concentration of $SO_2$ in urban Beijing gradually decreased since 2012 due to strict emission control in northern China. An air pollution prevention and control action plan was conducted since 2013 and it led to a reduction of the PM$_{2.5}$ mass concentration in north China plain by more than 25%. However, the reduction of $SO_2$ and fine particle emissions and concentrations did not significantly change the frequency of new particle formation events in urban Beijing (Li et al., 2020). Hence, it is unlikely that the new particle formation measured in urban Beijing was often affected by industrial emissions.

In addition to confirming the local formation of new particles using aerosol size spectrometers down to the cluster size, the temporal evolution of the measured size distribution is a key to distinguish new particle formation events from plume events and traffic emissions. The examples given in Iida et al. (2008) show the identification of new particle

formation events in the polluted environment, and similar criterions were also summarized in Kulmala et al. (2012) and Kerminen et al. (2018). Figure R2 shows a plume event measured in urban Beijing. At 10:00, new particles were observed at ~3 nm and these particles grew steadily to ~20 nm at 12:00, which followed the typical pattern of a new particle formation event. However, starting from 14:30, new particles ranging from ~1 nm to ~10 nm with a higher concentration than the existing new particles were observed. Since ~10 nm particles were observed almost simultaneously with ~1 nm particles at 14:30 and there was no clear growth pattern in the measured aerosol size distribution, this event during 14:30 – 17:30 was likely to be a plume event. Note that since new particle down to ~1 nm were observed during this plume event, the source of this plume should be close to rather than away from the sampling site. In contrast, the formation and growth pattern in Fig. 7a indicates that the event observed on Feb. 24, 2018 was a regional new particle formation event.

[Figure]

Figure R2. Aerosol size distributions measured in urban Beijing on Apr. 2, 2014.

**Minor comments**

Who is Julia (Page 7, line 1)

**Response**: We revised "Julia" as "the Julia programming language". Julia is the name of this programming language.

A reference to a paper under review (Cai et al, 2020) still has to be considered as grey literature as long as the review process is not open access. Also, as it is not clear whether the companion manuscript passes the review process, the information in this other paper is crucial for the current manuscript but might not be available. It could be placed in a supplement.

**Response**: We replaced Cai et al. (2020) with Deng et al. (2020b). Deng et al. (2020b) has been published and the details on the measurements are detailed therein.

A reference to a paper in preparation (Li and Cai, 2020) is not even grey literature!

**Response**: Li and Cai (2020) has been published now. The citation information is updated.

Define the conventional method, when it is first cited, Page 5, Lethipalo, 2014? Later in the paper it is confusing, with which model the current results are compared without a reference.

**Response**: Thanks. We added add a brief description of the conventional appearance time method in Lethipalo et al. (2014).

To clarify the used of the models, we added a paragraph at the beginning of Section 3. The introduction to the models in section 3.1 was revised as "A discrete aerosol model and a discrete-sectional aerosol model based on aerosol dynamics were used to provide an evolving aerosol size distribution and hence to test the conventional and corrected appearance time methods." Li and Cai (2020) is the reference for the discrete-sectional model, which also illustrates the principles of a discrete model. Note that these models were not compared to experiments, because they are used to generate an evolving size distribution rather than to investigate the nucleation and growth mechanism.

The same holds for the theory when the model, which is theory as well, is compared to another theory.

**Response**: The mechanism for the evolution of aerosol size distribution given by the model was not compared to other nucleation mechanisms. These aerosol dynamics models are used to provide growth rate in consistency with the simulated aerosol size distributions. Since the nucleation and growth mechanism used in these models do not affect this consistency, they do not affect the validity of these models in testing the formulae of the appearance time method. In Fig. 7b, the growth rate retrieved from the measured aerosol size distributions using the appearance time method is compared to the condensational growth rate contributed by $H_2SO_4$.

Within the time has to be defined in the text and figures. Is this Beijing CST?

**Response**: Yes. We revised the x-label in Fig. 7a as "Local time (HH:MM)". The time in Figs. 1 and 6 are the simulation time and it does not correspond to the time in a certain day. Hence, we keep the x-labels in Figs. 1 and 6.

Numbering of the tables. Why is the third table named table A1 instead of Table 3?

**Response**: We intend to put table A1 in the appendix. "A" is short for "appendix".

**References**

Cai, R., Yang, D., Fu, Y., Wang, X., Li, X., Ma, Y., Hao, J., Zheng, J., and Jiang, J.: Aerosol surface area concentration: a governing factor in new particle formation in Beijing, Atmospheric Chemistry and Physics, 17, 12327-12340, 10.5194/acp-17-12327-2017, 2017.

Deng, C., Cai, R., Yan, C., Zheng, J., and Jiang, J.: Formation and growth of sub-3 nm particles in megacities: impacts of background aerosols, Faraday Discussions, 10.1039/D0FD00083C, 2020a.

Deng, C., Fu, Y., Dada, L., Yan, C., Cai, R., Yang, D., Zhou, Y., Yin, R., Lu, Y., Li, X., Qiao, X., Fan, X., Nie, W., Kontkanen, J., Kangasluoma, J., Chu, B., Ding, A., Kerminen, V., Paasonen, P., Worsnop, R. D., Bianchi, F., Liu, Y., Zheng, J., Wang, L., Kulmala, M., and Jiang, J.: Seasonal characteristics of new particle formation and growth in urban Beijing, Environmental Science and Technology, 54, 8547–8557, 10.1021/acs.est.0c00808, 2020b.

Kenjiro, I., Stolzenburg, M. R., McMurry, P. H., and Smith J. N.: Estimating nanoparticle growth rates from size-dependent charged fractions: Analysis of new particle formation events in Mexico City, J. Geophys. Res., 113, D05207, doi:10.1029/2007JD009260, 2008.

Kuang, C., Chen, M., Zhao, J., Smith, J., McMurry, P. H., and Wang, J.: Size and time-resolved growth rate measurements of 1 to 5 nm freshly formed atmospheric nuclei, Atmospheric Chemistry and Physics, 12, 3573-3589, 10.5194/acp-12-3573-2012, 2012.

Kulmala, M., Vehkamäki, H., Petäjä, T., Dal Maso, M., Lauri, A., Kerminen, V.-M., Birmili, W., and McMurry, P. H.: Formation and growth rates of ultrafine atmospheric particles: a review of observations, Journal of Aerosol Science, 35, 143-176, 10.1016/j.jaerosci.2003.10.003, 2004.

Kulmala, M., Petäjä, T., Nieminen, T., Sipilä, M., Manninen, H. E., Lehtipalo, K., Dal Maso, M., Aalto, P. P., Junninen, H., Paasonen, P., Riipinen, I., Lehtinen, K. E., Laaksonen, A., and Kerminen, V.-M.: Measurement of the nucleation of atmospheric aerosol particles, Nature protocols, 7, 1651-1667, 10.1038/nprot.2012.091, 2012.

Kulmala, M., Kontkanen, J., Junninen, H., Lehtipalo, K., Manninen, H. E., Nieminen, T., Petäjä, T., Sipilä, M., Schobesberger, S., Rantala, P., Franchin, A., Jokinen, T., Jarvinen, E., Äijälä, M., Kangasluoma, J., Hakala, J., Aalto, P. P., Paasonen, P., Mikkilä, J., Vanhanen, J., Aalto, J., Hakola, H., Makkonen, U., Ruuskanen, T., Mauldin, R. L., 3rd, Duplissy, J., Vehkamäki, H., Bäck, J., Kortelainen, A., Riipinen, I., Kurtén, T., Johnston, M. V., Smith, J. N., Ehn, M., Mentel, T. F., Lehtinen, K. E., Laaksonen, A., Kerminen, V.-M., and Worsnop, D. R.: Direct observations of atmospheric aerosol nucleation, Science, 339, 943-946, 10.1126/science.1227385, 2013.

---

## Author Comment (AC2) · 11 Sep 2020

**Responses to Reviewer #2's Comments on Manuscript acp-2020-398**

(Impacts of coagulation on the appearance time method for sub-3nm particle growth rate evaluation and their corrections)

We thank the anonymous referee (referred as reviewer #2 below) for the efforts and constructive comments that help to improve this manuscript. The reviewer's comments are addressed in the following paragraphs and the manuscript were revised majorly. The comments are shown as sans-serif dark red texts and our responses are shown as serif black texts. Changes are highlighted in the revised manuscript and shown as "quoted underlined texts" in the responses. Line numbers, figures, and equations quoted in the responses correspond the revised manuscript. References are given at the end of the responses.

**Reviewer #2**

This manuscript investigates the impact of coagulation on the particle growth rate, calculated using the appearance time method, using theoretical derivations and aerosol dynamics modeling. The topic of actual growth rate calculation is of great importance for understanding the new particle formation processes in the atmosphere. The appearance time method was originally developed by Lehtipalo et al., 2014 to calculate growth rate in the size range 1-3nm using PSM data. They highlighted that the method is robust unless coagulation process affect greatly the particles size distributions, such as a heavily polluted environment with high number concentration of preexisting particles. Although this paper is presenting a correction for coagulation on the appearance time method, the approach and the validity of the method are not adequately described. It would be more appropriate to first describe the method and its weaknesses, then present the suggested correction for coagulation impacts and then apply the method in different environment types (boreal forest and Beijing data are available for comparison).These steps, are only briefly described and definitely more examples need to be presented.

**Response**: We agree with the reviewer that the limitations of the appearance time method should be discussed, though the majority of this manuscript is on the derivation and validation of the correction formula for the appearance time method. The appearance time method is described in the revised Section 2.3. We added section 4.5 on the uncertainties of the appearance time method and the challenges in determining particle growth rate in the atmosphere.

However, we prefer not to extend the scope of this manuscript to applying the appearance time methods to various types of new particle formation events. The impacts of coagulation to the appearance time in the Finnish boreal forest and urban Beijing are indicated in Fig. A1. Meanwhile, as explained in section 4.5, the validation of the appearance time method and other methods in the real atmospheres needs more systematic and comprehensive investigations.

**Specific comments:**

Page 2, Line 19: Only few or just one application?

**Response**: According to our knowledge, there is only one published application (Kuang et al., 2012) and some unpublished applications. This sentence was revised as "However, few applications……(e.g., Kuang et al., 2012)" so that is does not emphasize the exact number of applications.

Page 4, Line 13 and 25 and further: Coagulation coefficient unit should be $cm^3 \ s^{-1}$.

**Response**: Thanks. We checked the manuscript and corrected the typos.

Page 5, Line 24: As this paper describes a correction to the appearance time method it is proper to present the method.

**Response**: Thanks. We added a new paragraph at the beginning of this section to describe the appearance time method.

Page 6, Line 17: This section is lacking all the necessary information for the reader to understand the theoretical and experimental tools that were used to perform this study.

**Response**: We added a paragraph at the beginning of section 3, which introduces why the simulated and measured new particle formation events are used to test the formulae and what information can be obtained from these test.

**Response**: Julia is a programming language. We replaced "Julia" with "the Julia programming language" in the revised manuscript.

Page 7, Line 20: This reference is a paper under review. More details about the experimental part should be given here.

**Response**: We added the environmental information on during the presented new particle formation event. The paper under review was replaced by a published paper (Deng et al., 2020b), in which the measurements are detailed.

Page 7, Line 23: This section (4.1) needs to be moved to methods, where the appearance method should be described and cited.

**Response**: We prefer to keep this section here because this section is mainly on the derivation of the appearance time method under an ideal condition. The derivation itself is a result. Instead, we added a paragraph in 2.3 to introduce the appearance time method and refer to readers to Section 4.1 for more details.

Page 11, Line 14: This is not shown here, we have no indication about sub-1.3nm growth rates.

**Response**: The reviewer mistook Fig. 5 for Fig. 4. The size range for particle growth rate in Fig. 4 covers 1.0 – 3.5 nm.

Page 11, Line 15:In Figure 4, Coags corrected seems to perform better than the Corrected total growth for particles larger than 3nm.

**Response**: Yes. This is because the correction term for CoagSrc is an approximation which cannot avoid potential bias. These uncertainties are discussed at the end of Section 4.2. However, considering uncertainties, Fig. 5 (instead of Fig. 4) shows that both the CoagS corrected growth rate and the corrected total growth rate agree with the theoretical growth rate.

Page 11, Line 26: This paragraph is confusing. Which formula is used, Eq 5, or some other formula from the Appendix?

**Response**: We added "These three impacts of coagulation source are accounted for in the corrected appearance time method (Eqs. 6 and 7)".

Page 12, Line 4: It has to be shown here that the method is described so that all limitations are discussed prior to applying the new method.

**Response**: The limitation of the correction formulae (Eq. 6) is that the coagulation source term is derivated based on approximation. This limitation has been clarified during the derivation and in this paragraph.

Page 12, Line 13: It is the previous study, or are there more studies?

**Response**: We added another reference (Olenius et al., 2014) to this sentence.

Page 12, Line 24: I would not use the expression agrees better, as it does not seem to agree, it seems to work better than the conventional method but still overestimates all particle growth rates outside the range 2-3.5 nm. These discrepancies both in absolute values but also with regard to increasing size and especially the shape of the curve have to be discussed further in the text. It has to be noted that the new curve has the same shape as the uncorrected one which suggests that there is an underlying assumption causing these deviations, it is worth providing more information.

**Response**: We revised this sentence as "the deviation between the corrected particle growth rate and the theoretical growth rate is smaller than the deviation between the conventional growth rate and the theoretical growth rate". The discrepancy outsize the range 2-3.5 nm has been emphasized in the next sentence: "However, for particles larger than ~5 nm and smaller than ~2 nm, the appearance time method overestimates particle growth rate even after correcting the impact of coagulation in the test case." We also added "These overestimations are caused by approximating the influence of coagulation source on particle growth with the coagulation source term in Eq. 5" to illustrate the reason for these discrepancies. In Section 4.2, we have clarified the reason why we use the approximation though it may introduce bias: "Since CoagSrc$_i$ is determined by the concentrations of all particles containing 2 to i-2 molecules, it is difficult to obtain an analytical solution of Eq. 19 without approximation. Here we provide an approximation method to correct the impact of coagulation source to the appearance time."

Page 12, Line 34: This assumption is valid for cases with clear diurnal variations of vapor concentrations as the assumed one. However what happens when condensing species exist in the afternoon as well, then the GR would be much higher. The example in 4.4 is demonstrating this weakness as in the afternoon the GR calculation is three time higher. Sensitivity tests with condensing species not vanishing in the afternoon could be useful as well.

**Response**: We add "To reduce this systematic bias due to a decreasing vapor concentration…" to emphasize the cause of this bias. If the vapor concentration stays relatively stable after a certain moment, the discrepancy between the theoretical growth rate and the growth rate retrieved using the appearance time method should be smaller. Fig. 5 shows a smaller discrepancy under a constant monomer concertation.

The reviewer misunderstood the example in Section 4.4. In Figs. 2-6, there is a theoretical growth rate given by the model that can be taken as a reference for the retrieved growth rate using the appearance time method. In contrast, the aerosol size distributions in Fig. 7 in Section 4.4 were measured in the real atmosphere and the true growth rate is unknown. Hence, there is no reference growth rate to justify whether the growth rate retrieved using the appearance time method in Fig. 7 is underestimated or overestimated. The condensational growth rate in Fig. 7 was estimated using the measured $H_2SO_4$ concentration, which is not the true growth rate. We have clarified in Section 4.4 that "Note that the sum of theoretical condensation and coagulation growth rate is not necessarily equal to the theoretical total growth rate for the measured NPF event. This is because only the condensation of sulfuric acid is considered whereas other vapors may also contribute to new particle growth." As a result, condensation of other vapors on the grown particles is more like to be the reason for that the growth rate retrieved using the appearance time method is ~3 times larger than the condensational growth rate calculated using $H_2SO_4$ concentration. In the main text, we have clarified that "The deviation between the measured growth and theoretical growth for particles larger than ~3 nm indicates that there are other chemical species in addition to sulfuric acid (and the bases to neutralize it) contributing to particle growth."

Page 13, Line 4: A single NPF event is not enough to demonstrate the validity of the proposed correction. Different events, under various meteorological and environmental conditions and under different environment types (and hence condensing species) are necessary to my opinion to test the new formulae.

**Response**: As clarified in the above response, tests using measured new particle formation events does not validate the proposed formula because the true growth rate is unknown. Instead, the validity and limitations of the proposed formula are tested using the models in the above sections. The aim of this test using a measured new particle formation event is to give an intuitive example of the impacts of coagulation to the appearance time method, though such impacts have been theoretically predicted in Fig. A1. An example using more new particle formation events is given in Deng et al. (2020a) to shown the comparison between the conventional and appearance time method, yet such a comparison is not a validation.

We added "In addition to the example given in Fig. 7, the average growth rates of 1.5-3 nm particles measured in urban Beijing retrieved using the conventional and corrected appearance time methods were reported in Deng et al. (2020a)." in section 4.4.

We also added a new section 4.5 on the uncertainties of the appearance time method in atmospheric application. As discussed therein, growth rate calculation is usually sensitive to uncertainties. A compressive study including more applications in the atmosphere, comparison between the results retrieved using different methods, and theoretical study based on simulation with uncertainties are needed to figure out a more accurate and robust method for growth rate estimation in the atmosphere.

**References**

Deng, C., Cai, R., Yan, C., Zheng, J., and Jiang, J.: Formation and growth of sub-3 nm particles in megacities: impacts of background aerosols, Faraday Discussions, 10.1039/D0FD00083C, 2020a.

Deng, C., Fu, Y., Dada, L., Yan, C., Cai, R., Yang, D., Zhou, Y., Yin, R., Lu, Y., Li, X., Qiao, X., Fan, X., Nie, W., Kontkanen, J., Kangasluoma, J., Chu, B., Ding, A., Kerminen, V., Paasonen, P., Worsnop, R. D., Bianchi, F., Liu, Y., Zheng, J., Wang, L., Kulmala, M., and Jiang, J.: Seasonal characteristics of new particle formation and growth in urban Beijing, Environmental Science and Technology, 54, 8547–8557, 10.1021/acs.est.0c00808, 2020b.

Kuang, C., Chen, M., Zhao, J., Smith, J., McMurry, P. H., and Wang, J.: Size and time-resolved growth rate measurements of 1 to 5 nm freshly formed atmospheric nuclei, Atmospheric Chemistry and Physics, 12, 3573-3589, 10.5194/acp-12-3573-2012, 2012.

Olenius, T., Riipinen, I., Lehtipalo, K., and Vehkamäki, H.: Growth rates of atmospheric molecular clusters based on appearance times and collision–evaporation fluxes: Growth by monomers, Journal of Aerosol Science, 78, 55-70, 10.1016/j.jaerosci.2014.08.008, 2014.

---

## Editor Comment (EC1) · Radovan Krejci (Editor) · 23 Oct 2020

Review of "Impacts of coagulation on the appearance time method for sub-3 nm particle growth rate evaluation and their corrections" by Cai et al.

The manuscript presents an attempt to correct apparent nanoparticle growth rates, obtained by the so-called appearance time method, to remove errors arising from co-agulation. It is true that the appearance time method may cause significant artefacts in the experimentally deduced growth rate and consequently in its interpretation and comparisons with models. The purpose of the work, i.e. to test the derived corrections against a particle dynamics model (rather than to compare the used models to mea-

surements), is clarified in the revised manuscript. However, my main concerns are that (a) it seems that the suggested corrections do not generally work, and (b) the modeled test cases are quite limited, and for example assume that sub-3 nm particles do not evaporate which is quite a restrictive assumption. Below are the detailed comments that I would ask the authors to address:

1. Figure 6 does not look very convincing in terms of the performance of the suggested correction approaches – do other test cases exhibit similar behavior in the size-dependent errors? Rather than listing the average discrepancies in the corrected growth rates e.g. in the Abstract and Conclusions, the maximum errors should be stated, since they are up to 150 % (!) for the present test cases. In general, complex particle population dynamics cannot normally be described by simplifications, so it is not so surprising that the corrections do not work very well, or cannot be reliably applied on realistic data. This is one of the main results of the work and should be highlighted – even a negative result is a result. Instead of stating "the feasibility of the corrected method was verified" it can be said that the method was "tested", since based on the presented results it doesn't look like the method was verified. Also the title is a bit misleading since it talks about sub-3 nm particles, while later in the test it's concluded that neither the conventional nor the corrected method work reliably at sub-2 nm. I would therefore reformulate the essential parts of the text to discuss "suggestions" or "attempts" to correct, instead of "corrections".

2. The omission of evaporation in the simulations limits the applicability of the results, since atmospheric small clusters and particles generally evaporate significantly. (A single test with a size-independent evaporation rate doesn't help very much, since for the smallest particles the rate may vary even by orders of magnitude with the size and composition, and the used rate is also quite low.) Therefore, I'd ask to include test cases where the smallest clusters have strongly size-dependent evaporation rates of at least around 10^-2 s^-1, or even higher. For example the highest values in Schobesberger et al., Atmos. Chem. Phys. 15, 55-78, 2015 could be used for upper-limit

estimates for the evaporation effects. Specifically, since often the evaporation rates of the smallest clusters exceed the vapor condensation rate, it's essential to include cases where $\beta^*N1$-E (Sect. 4.1, page 9) is negative. Stating that "the bias caused by this size dependency of evaporation is similar to that of coagulation" is odd, since evaporation and coagulation processes are very different: the former moves particles along the size axis within the studied size range, while the latter removes particles from the size range. Can you show how their effects are similar (for arbitrary evaporation and coagulation rate constants)? In Figure 2, how would a test case with both non-zero sink and non-zero evaporation look like? The effects of evaporation may be different at sinks of different magnitude.

3. Sect. 4.2: it is concluded that CoagSrc does not have a major impact on the apparent growth rate. This statement could be softened, since the test cases are so limited, and also the vapor concentrations here are not extremely high. Maybe CoagSrc may still have a larger role at the higher end of atmospherically relevant vapor concentrations, e.g. at ca. $10^8$ cmˆ-3?

4. The particle size is mainly expressed through the diameter. The results should thus be dependent on the size of the vapor molecules: the diameters of very small nanoparticles containing equal numbers of molecules are different for e.g. large organic molecules and sulfuric acid. Also, the behavior of apparent GR with respect to diameter may be different for multi-component particle formation where the sizes of the molecular species differ significantly. Which molecular size was used in the simulations? In Sect. 4.3 it is stated that "one should be cautious about the sub-2 nm size-resolved growth rate" – could the 2-nm-limit be something else for different molecules, for example could the diameter threshold be larger for highly oxidized organic monomers or dimers? I recommend to bring up the fact that the diameters may not be very meaningful for such small clusters.

5. Derivation of the corrections in the Appendix: Can you elaborate how the first very first equality (GRconv = ...) in Eq. (A9) is deduced; it is not obvious. Also, in the

derivation of Eq. (A6, it is assumed that the concentration at the appearance time is 50 % of the final value. It should be noted that in the "conventional" appearance time method it is actually defined that the appearance time concentration is 50% of the maximum value (Lehtipalo et al., 2014), which is not the same as the final value in a strong clustering event. This can affect even "ideal" cases with a constant vapor concentration. In general, the correction seems to be derived for a situation where the particle distribution relaxes into a steady state (equations in the Appendix with $N_{i,inf}$). Thus I don't see a reason to believe that the appearance time method, or any "corrected" version of it, would work for realistic, dynamic atmospheric environments with varying vapor and particle concentrations, so the failure in Fig. 6 is not surprising. Why could e.g. Eq. (A8) be applied to an atmospheric non-steady-state situation?

Minor comments: 1. After Eq. (1): "Note that Eq. 1 is expressed in the discrete form, i.e., it does not assume a continuum particle size": one can note that it does however assume a well-defined GR typical for the continuum space, i.e. no spreading of the size distribution, and no negative average net condensation flux in the Lagrangian presentation (see comment 2). 2. Sect. 4.1: Has the appearance time method actually been derived in the original paper (Lehtipalo et al., 2014), or is this something done only in the present manuscript? Is e.g. the original choice of 50 % concentration increase arbitrary? 3. Table A1 and Figure 6: Would a varying coagulation sink affect the results? The growth of the boundary layer in daytime typically causes a time-dependent sink.

---

## Author Comment (AC3) · 23 Nov 2020

**Responses to Reviewer #3's Comments on Manuscript acp-2020-398**

(Impacts of coagulation on the appearance time method for sub-3nm particle growth rate evaluation and their corrections)

We thank the editor, Dr. Radovan Krejci (referred as reviewer #3 below) for the deep insights and constructive comments that help to improve this manuscript. The reviewer's comments are addressed in the following paragraphs and the manuscript were revised majorly. In addition to the influences of coagulation on the appearance time method and their corrections, we discuss the uncertainties of the conventional and corrected appearance time method. The potential biases of the corrected appearance time method due to vapor evaporation and the varying vapor concentration are reported and discussed. More simulation results are presented to support these discussions. Some of these simulation results are included as supplementary information. The comments are shown as sans-serif dark red texts and our responses are shown as serif black texts. Changes are highlighted in the revised manuscript and shown as "quoted underlined texts" in the responses. Line numbers, figures, and equations quoted in the responses correspond the revised manuscript. References are given at the end of the responses.

**Reviewer #3**

The manuscript presents an attempt to correct apparent nanoparticle growth rates, obtained by the so-called appearance time method, to remove errors arising from coagulation. It is true that the appearance time method may cause significant artefacts in the experimentally deduced growth rate and consequently in its interpretation and comparisons with models. The purpose of the work, i.e. to test the derived corrections against a particle dynamics model (rather than to compare the used models to measurements), is clarified in the revised manuscript. However, my main concerns are that (a) it seems that the suggested corrections do not generally work, and (b) the modeled test cases are quite limited, and for example assume that sub-3 nm particles do not evaporate which is quite a restrictive assumption.

**Response**: We thank the reviewer for these comments. The appearance time method was proposed to estimate the growth rate of clusters and new particles. As summarized in the Introduction, it is usually used in the sub-3 nm size range because other methods are difficult to cover this size range. 3 nm here is a rough estimation rather than a critical threshold. Previously studies have reported considerable potential uncertainties of the appearance time method (Olenius et al., 2014; Kontkanen et al., 2016; Li and McMurry, 2018), yet such potential uncertainties were usually not accounted for when using the appearance time method. The theoretical basis for the appearance time method under an ideal condition (constant vapor source, no coagulation, and no external sink) was reported quite recently (He et al., 2020). Based on derivations, this manuscript shows that coagulation has impacts on the appearance time and these impacts can be corrected using the measured aerosol size distributions. The corrections are validated by the derivations and elaborated in Figs. 1-5. However, the proposed corrections are only for the influences of coagulation. For instance, vapor evaporation and the variation of vapor concentration are not accounted for in the corrected formula. Hence, there are still uncertainties in the corrected formula. Reporting these uncertainties is one of the contributions of the revised manuscript. These revisions are illustrated and discussed in detail in the responses below.

In response to the two specific main concerns: a) The correction for influences of coagulation has been validated by derivations and illustrated using test results. In the revised manuscript, we distinguish "the correction for coagulation influence" from "the corrected appearance time method". We agree with the reviewer that there may be uncertainties in

the corrected formula, yet the test results show that correcting the influences coagulation reduces these uncertainties. b) We added the discussions on vapor evaporation to the revised manuscript.

Below are the detailed comments that I would ask the authors to address:

1. Figure 6 does not look very convincing in terms of the performance of the suggested correction approaches – do other test cases exhibit similar behavior in the size-dependent errors? Rather than listing the average discrepancies in the corrected growth rates e.g. in the Abstract and Conclusions, the maximum errors should be stated, since they are up to 150 % (!) for the present test cases. In general, complex particle population dynamics cannot normally be described by simplifications, so it is not so surprising that the corrections do not work very well, or cannot be reliably applied on realistic data. This is one of the main results of the work and should be highlighted – even a negative result is a result. Instead of stating "the feasibility of the corrected method was verified" it can be said that the method was "tested", since based on the presented results it doesn't look like the method was verified. Also the title is a bit misleading since it talks about sub-3 nm particles, while later in the test it's concluded that neither the conventional nor the corrected method work reliably at sub-2 nm. I would therefore reformulate the essential parts of the text to discuss "suggestions" or "attempts" to correct, instead of "corrections".

**Response**: In the revised manuscript, we revised the statements such that correction and validation are only used for the influences of coagulation. Meanwhile, we emphasized that although this correction reduces the biases of the appearance time method, there are still uncertainties in the growth rate estimated using the appearance time method due to vapor evaporation and the variation of vapor concentration. These uncertainties are indicated by simulation results and reported in the abstract and conclusions.

Figure 7 (the original Fig. 6) shows the typical trend of size-dependent error of the appearance time method, whereas the amount of error varies with simulation conditions and it is indicated in Table A1. After correcting the influences of coagulation, the bias of the appearance time method is reduced.

We report the maximum errors of the size-dependent growth rate of the simulation results in the revised Abstract and Conclusions. We agree with the reviewer of 150% is a huge error; however, the uncertainties in measurements of sub-10 nm aerosol size distribution often exceeds this value (Kangasluoma et al., 2020). These uncertainties will propagate if the absolute aerosol concentration is used to estimate the growth rate, whereas the appearance time method uses only the variation of aerosol size distribution.

"Sub-3 nm particle" in the title and abstract was revised as "new particle".

In summary, the revised manuscript emphasizes more on the uncertainties of the appearance time method. We limit the correction and validation to the influences of coagulation and clarified that there are remaining uncertainties in the appearance time method. Reporting these uncertainties is a contribution of this study.

2. The omission of evaporation in the simulations limits the applicability of the results, since atmospheric small clusters and particles generally evaporate significantly. (A single test with a size-independent evaporation rate doesn't help very much, since for the smallest particles the rate may vary even by orders of magnitude with the size and composition, and the used rate is also quite low.) Therefore, I'd ask to include test cases where the smallest clusters have strongly size-dependent evaporation rates of at least around $10^{-2}$ s$^{-1}$, or even higher. For example the highest values in Schobesberger et al., Atmos. Chem. Phys. 15, 55-78, 2015 could be used for upper-limit estimates for the evaporation effects. Specifically, since often the evaporation rates of the smallest clusters exceed the vapor condensation rate, it's essential to include cases where $\beta*N_1-E$ (Sect. 4.1, page 9) is negative. Stating that "the bias caused by this size dependency of evaporation is similar to that of coagulation" is odd, since evaporation and coagulation processes are very different: the former moves particles along the size axis within the studied size range, while the latter removes particles from the size

range. Can you show how their effects are similar (for arbitrary evaporation and coagulation rate constants)? In Figure 2, how would a test case with both non-zero sink and non-zero evaporation look like? The effects of evaporation may be different at sinks of different magnitude.

**Response**: We added discussions on the influences of vapor evaporation together with a new Fig. 6 to section 4.3. Size-dependent evaporation was assumed for the simulation in Fig. 6. The evaporation rate of the smallest cluster (dimer) was assumed to be ~0.2 s$^{-1}$ according to the quantum chemistry results for $H_2SO_4$-$NH_3$ nucleation (Myllys et al., 2019). For sub-1.5 nm particles, the net flux for monomer condensation in the Lagrangian specification ($\beta N_1$-$E$) is negative. The simulation results show that in addition to changing the growth rate, evaporation may also influence the steady-state concentration of particles and hence impact the retrieved growth rate. With prior information on the size-dependent evaporation rate, this influence can be readily corrected and the corrected net condensation growth rate agrees with the theoretical value. However, since the size-dependent evaporation rate is rarely known, this correction may be not available when applying the appearance time method. Neglecting the influence of vapor evaporation on the appearance time causes an overestimation of the growth rate smaller or slightly larger than the critical size (at which $\beta N_1$-$E = 0$). This finding is supported by tests with different size-dependent evaporation rates. Hence, with a correction for vapor evaporation, the appearance time method is not valid to characterize net particle/cluster growth during nucleation ($\beta N_1$-$E < 0$). In Conclusions, we added "Further, the growth rate of vapors and clusters is recommended to be estimated based on cluster dynamics instead of their representative time."

"The bias caused by this size dependency of evaporation is similar to that of coagulation" in the original manuscript means refers to similarities in their mathematical expressions and corrections. Figure 6 shows that the correction for evaporation is similar to that for coagulation, i.e., they reduce the steady-state concentration of particles and can be corrected similarly. However, this sentence was removed to avoid confusion.

In addition to the influence of evaporation for a homo-molecular nucleation system, the supporting information includes a test with a volatile vapor and a non-volatile vapor. The appearance time method follows the theoretical growth rate in this test.

3. Sect. 4.2: it is concluded that CoagSrc does not have a major impact on the apparent growth rate. This statement could be softened, since the test cases are so limited, and also the vapor concentrations here are not extremely high. Maybe CoagSrc may still have a larger role at the higher end of atmospherically relevant vapor concentrations, e.g. at ca. 10ˆ8 cmˆ-3?

**Response**: This sentence was revised as "CoagSrc does not have a major impact on the apparent growth rate of sub-10 nm particles even during an intensive atmospheric NPF event in urban Beijing." We agree with the reviewer that the coagulation source may play an important role in new particle growth in some atmospheric environments. However, a high vapor concentration does not necessarily correspond to a high coagulation source because the coagulation source is determined by the concentration of new particles.

4. The particle size is mainly expressed through the diameter. The results should thus be dependent on the size of the vapor molecules: the diameters of very small nanoparticles containing equal numbers of molecules are different for e.g. large organic molecules and sulfuric acid. Also, the behavior of apparent GR with respect to diameter may be different for multi-component particle formation where the sizes of the molecular species differ significantly. Which molecular size was used in the simulations? In Sect. 4.3 it is stated that "one should be cautious about the sub-2 nm size-resolved growth rate" – could the 2-nm-limit be something else for different molecules, for example could the diameter threshold

be larger for highly oxidized organic monomers or dimers? I recommend to bring up the fact that the diameters may not be very meaningful for such small clusters.

**Response**: We clarify that the vapor for simulation is sulfuric acid in the revised section 3.1. A new simulation with large molecules (400 Da) was added to the supporting information. In the revised manuscript, we added "For particles close to the size of vapor molecules (sub-2 nm in these tests), the appearance time usually convolves other information (e.g., the varying vapor concentration and the size-dependent coagulation coefficient) in addition to particle growth. Figure S2 shows that with larger vapor molecules, the size range for the discrepancy between the theoretical and retrieved growth rate shifts towards larger diameters."

5. Derivation of the corrections in the Appendix: Can you elaborate how the first very first equality (GRconv = . . .) in Eq. (A9) is deduced; it is not obvious. Also, in the derivation of Eq. (A6, it is assumed that the concentration at the appearance time is 50 % of the final value. It should be noted that in the "conventional" appearance time method it is actually defined that the appearance time concentration is 50% of the maximum value (Lehtipalo et al., 2014), which is not the same as the final value in a strong clustering event. This can affect even "ideal" cases with a constant vapor concentration. In general, the correction seems to be derived for a situation where the particle distribution relaxes into a steady state (equations in the Appendix with N_i,inf). Thus I don't see a reason to believe that the appearance time method, or any "corrected" version of it, would work for realistic, dynamic atmospheric environments with varying vapor and particle concentrations, so the failure in Fig. 6 is not surprising. Why could e.g. Eq. (A8) be applied to an atmospheric non-steady-state situation?

**Response**: We added a sentence to illustrate the first line in Eq. S6 (original Eq.A9): "The appearance time and hence the retrieved growth rate are mainly influenced in two aspects: 1) the steady-state concentration and 2) the particle source that determines the time to reach a certain steady-state concentration."

As clarified in Section 4.1 and the recently published He et al. (2020), the appearance time should be defined with respect to the steady-state concentration, which is also equal to its maximum concentration if all the assumptions for derivation are valid. We agree with the reviewer that in the conventional appearance time method, the appearance time is usually calculated using the maximum concentration. However, this should be taken as an approximation rather than a definition. In section 4.3, we added "The 50% appearance time is herein calculated using the maximum size-resolved particle concentration because $N_{i,\infty}$ is not available, and this approximation introduces biases to the retrieved growth rate."

We also agree with the reviewer that the appearance time methods and its correction are derived based on an ideal assumption of constant vapor concentration and constant sink, which is usually not valid for the real atmosphere. Hence, we listed the assumptions for the derivations of the appearance time method at the very beginning of section 4.1. In the revised manuscript, we also add a new paragraph to emphasize the foreseeable uncertainties of the appearance time method due to the violation of these ideal assumptions: "It can be seen that none of these ideal conditions is consistent with real atmospheric environments. Since the conventional appearance time method is derived based on these conditions, violating them may cause biases in the appearance time method. We will first show brief derivations of the conventional appearance time and then discuss the correction for the influences of coagulation and other remaining potential uncertainties of the corrected appearance time method."

We disagree with the reviewer on the statement that the appearance time method does not work for the real atmosphere. It is true that due to the variation of vapor concentration, the appearance time can never be strictly accurate for new particle formation in the real atmosphere. For instance, applying Eq. S5 to an atmospheric non-steady-state situation is an approximation and it leads to uncertainties in the retrieved growth rate. Figure 7 and the data in Table A1

indicate these uncertainties caused by a varying vapor concentration. However, the appearance time method can report a growth rate close to the theoretical value (although with large uncertainties) and the correction for the influences of coagulation reduces these uncertainties. Considering the fact that other methods (representative diameter methods and the methods based on solving aerosol general dynamic equations) can rarely report a growth rate for sub-5 nm particles and the large uncertainty in determining the absolute concentration of sub-10 nm particles (up to a factor of 10, Kangasluoma et al., 2020), we think that the uncertainties of the corrected appearance time method shown in Fig. 7 are acceptable.

As clarified in the Introduction, the appearance time method is favored for sub-3 nm particles because of its above advantages over other methods. Previous studies have reported potentially huge uncertainties (Olenius et al., 2014; Kontkanen et al., 2016; Li and McMurry, 2018) in the appearance time method, yet the appearance time method, and even the maximum concentration method, are still used for the real atmosphere and the uncertainties in the estimated growth rate are usually not addressed. Instead of simply reporting uncertainties, this study aims to 1) clarify the reason for these uncertainties, e.g., how evaporation may influence the retrieved growth rate and which size range is affected instead of reporting that evaporation may cause uncertainties, and 2) provide correction formulae for the influences of coagulation which reduces the uncertainties of the appearance time method.

**Minor comments:**

1. After Eq. (1): "Note that Eq. 1 is expressed in the discrete form, i.e., it does not assume a continuum particle size": one can note that it does however assume a well-defined GR typical for the continuum space, i.e. no spreading of the size distribution, and no negative average net condensation flux in the Lagrangian presentation (see comment 2).

**Response**: We revised the description of Eq. 1 as "When there is only one non-volatile condensing vapor……" and added "When considering particle evaporation, i.e., monomer dissociation, particle growth due to the net effect of monomer association and dissociation will be explicitly referred as net condensation growth" to address the cases with vapor evaporation. However, Eq. 1 is also valid for a spreading size distribution, as illustrated in Appendix A.

2. Sect. 4.1: Has the appearance time method actually been derived in the original paper (Lehtipalo et al., 2014), or is this something done only in the present manuscript? Is e.g. the original choice of 50 % concentration increase arbitrary?

**Response**: The derivation of the appearance time method under ideal conditions was reported quite recently (He et al., 2020). We include this reference in the revised manuscript. The threshold of 50 % was recommended by Lehtipalo et al. (2014) according to comparisons among simulation results. Section 4.1 does not discuss much on the usage of the appearance time method in previous studies because we think explaining the theory of the appearance time methods via derivations is more convincing than reviewing how it is used previously.

3. Table A1 and Figure 6: Would a varying coagulation sink affect the results? The growth of the boundary layer in daytime typically causes a time dependent sink.

**Response**: Coagulation sink influences the steady-state concentration of particles and its variation may introduce uncertainties to the retrieved growth rate. In the revised manuscript, we added "CoagS is assumed to be independent of time in the above discussions, whereas it may vary significantly during an NPF event in the atmosphere. The varying CoagS influences $N_{i,\infty}$ and hence the appearance time. Figure S3 shows that a bias of the growth rate retrieved using the appearance time method caused by a varying CoagS." Simulation results with a varying coagulation sink are presented in Fig. S3.

**Reference**

He, X.-C., Iyer, S., Sipilä, M., Ylisirniö, A., Peltola, M., Kontkanen, J., Baalbaki, R., Simon, M., Kürten, A., Tham, Y. J., Pesonen, J., Ahonen, L. R., Amanatidis, S., Amorim, A., Baccarini, A., Beck, L., Bianchi, F., Brilke, S., Chen, D., Chiu, R., Curtius, J., Dada, L., Dias, A., Dommen, J., Donahue, N. M., Duplissy, J., El Haddad, I., Finkenzeller, H., Fischer, L., Heinritzi, M., Hofbauer, V., Kangasluoma, J., Kim, C., Koenig, T. K., Kubečka, J., Kvashnin, A., Lamkaddam, H., Lee, C. P., Leiminger, M., Li, Z., Makhmutov, V., Xiao, M., Marten, R., Nie, W., Onnela, A., Partoll, E., Petäjä, T., Salo, V.-T., Schuchmann, S., Steiner, G., Stolzenburg, D., Stozhkov, Y., Tauber, C., Tomé, A., Väisänen, O., Vazquez-Pufleau, M., Volkamer, R., Wagner, A. C., Wang, M., Wang, Y., Wimmer, D., Winkler, P. M., Worsnop, D. R., Wu, Y., Yan, C., Ye, Q., Lehtinen, K., Nieminen, T., Manninen, H. E., Rissanen, M., Schobesberger, S., Lehtipalo, K., Baltensperger, U., Hansel, A., Kerminen, V.-M., Flagan, R. C., Kirkby, J., Kurtén, T., and Kulmala, M.: Determination of the Collision Rate Coefficient between Charged Iodic Acid Clusters and Iodic Acid using the Appearance Time Method, Aerosol Science and Technology, 1-17, 10.1080/02786826.2020.1839013, 2020.

Kangasluoma, J., Cai, R., Jiang, J., Deng, C., Stolzenburg, D., Ahonen, L. R., Chan, T., Fu, Y., Kim, C., Laurila, T. M., Zhou, Y., Dada, L., Sulo, J., Flagan, R. C., Kulmala, M., Petäjä, T., and Lehtinen, K.: Overview of measurements and current instrumentation for 1-10 nm aerosol particle number size distributions, Journal of Aerosol Science, 148, 105584, 10.1016/j.jaerosci.2020.105584, 2020.

Kontkanen, J., Olenius, T., Lehtipalo, K., Vehkamäki, H., Kulmala, M., and Lehtinen, K. E. J.: Growth of atmospheric clusters involving cluster–cluster collisions: comparison of different growth rate methods, Atmospheric Chemistry and Physics, 16, 5545-5560, 10.5194/acp-16-5545-2016, 2016.

Lehtipalo, K., Leppä, J., Kontkanen, J., Kangasluoma, J., Franchin, A., Wimmer, D., Schobesberger, S., Junninen, H., Petäjä, T., Sipilä, M., Mikkilä, J., Vanhanen, J., Worsnop, D. R., and Kulmala, M.: Methods for determining particle size distribution and growth rates between 1 and 3 nm using the Particle Size Magnifier, Boreal Environment Research, 19, 215-236, 2014.

Li, C., and McMurry, P. H.: Errors in nanoparticle growth rates inferred from measurements in chemically reacting aerosol systems, Atmospheric Chemistry and Physics, 18, 8979-8993, 10.5194/acp-18-8979-2018, 2018.

Myllys, N., Kubečka, J., Besel, V., Alfaouri, D., Olenius, T., Smith, J. N., and Passananti, M.: Role of base strength, cluster structure and charge in sulfuric-acid-driven particle formation, Atmospheric Chemistry and Physics, 19, 9753-9768, 10.5194/acp-19-9753-2019, 2019.

Olenius, T., Riipinen, I., Lehtipalo, K., and Vehkamäki, H.: Growth rates of atmospheric molecular clusters based on appearance times and collision–evaporation fluxes: Growth by monomers, Journal of Aerosol Science, 78, 55-70, 10.1016/j.jaerosci.2014.08.008, 2014.

---

## Referee Report (RR1)

2nd review of the manuscript

Impacts of coagulation on the appearance time method for sub-3 nm particle growth rate evaluation and their corrections

Author(s): Runlong Cai et al.

MS No.: acp-2020-398

The manuscript describes corrections for the calculation of growth rates after gas to particle conversion in an aerosol nucleation and growth model. Compared to the 'standard' model the initial growth with the new corrections are smaller than in the original 'conventional' calculation method.

The corrected growth rates are than compared to one day of ambient measurements in Beijing in Winter 2018.

The authors made some corrections to the manuscript to prevent misunderstandings and added and claim in a summary:

   We clarified that the models in this study were used to test the proposed formulae for the appearance time method. There is no comparison between the modeling results and measurements in this study. Besides, we clarified the input background aerosol concentration for the model in the revised manuscript.

*Fig. 7 shows a comparison of model results and measured ambient data. The input background aerosol concentration is given in a wide range of one order of magnitude. Due to the rapid change during the day a diurnal variation would be necessary.*

   We added more details on the experiments.

*The reference Deng et al, 2020b mentions the weather sensor, its location would be necessary. Also needed are diurnal variations of a complete set of meteorological variables.*

   We added a paragraph on the classification of the new particle formation event shown in Fig.7. The continuous formation of aerosols down to the cluster size and their subsequent growth are in accordance with the features of a regional new particle formation event rather than those of a plume event.

*The classification of a new particle formation event is not in agreement with a sulphur dioxide plume event (see Fig. R1).*

   The size-dependency of particle growth rate in the observed new particle formation event is similar to those reported in previous studies.

*Yes, but, this is not a proof of concept. There are many more nanoparticle events (NPE), classified as NPF but, clearly traceable to primary particle plumes.*

   The appearance time of H2SO4 and new particles on Feb. 24, 2018 indicates that the observed intensive new particle formation was mainly driven by daytime photochemical reaction.

*Daytime photochemical reactions are temporally not in agreement with the SO2 and H2SO4 diurnal pattern in Fig. 7 and Fig. R1, see for example Junkermann et al, 1989.*

*The question, which photochemical reaction is assumed to be responsible for the OH radical and H2SO4 production was not answered.*

Further comments

While the coagulation description is understandable, still the model description is marginal and does not allow a decision whether the model for the nucleation and growth rate calculation is applicable at all for a comparison with experimental data from the real atmosphere. It should be discussed under consideration of the limits of particle size distribution development in polluted atmospheres described by Kulmala et al (2017).

However:

It's well known since Aitken's (1890) work that horizontal transport and vertical mixing are the dominate factors controlling nanoparticle number concentrations. It's also known that under certain conditions, mainly in sulphur rich plumes, gas to particle conversion takes place (Mohnen and Lodge 1969, Stevens et al, 2012).

Thus, it is clear that during advection of sulphur rich air as shown in Fig R1 a rapid gas to particle conversion and nanoparticles in all size ranges can be expected. Whether these nanoparticles survive a few minutes or hours during transport and grow or shrink is a matter of the environmental conditions and the presence of other, especially larger, particles as recently discussed by Kulmala et al, (2017) , Kerminen et al, (2018), and Deng et al, (2020a). A complete description of these is conditions and their diurnal variation is required.

As mentioned by Deng et al, (2020b) also other local sources contribute (Rönkkö et al, 2017). For an example of nanoparticle transport from single elevated primary particle sources see Junkermann and Hacker (2018) and literature cited therein. For the distribution of such potential large sulphur, ammonia and nanoparticle sources affecting Beijing air pollution see for example www.endcoal.org.

Minor comments:

The argument that emission control and reduction of SO2 in China together with no corresponding change in NPE frequency is not an argument against industrial nanoparticle sources (see Junkermann et al, 2011). The claimed link between SO2 and H2SO4 peaks and the appearance of 1.5 nm particles on February 24 is even contradicting this argument.

It is not necessary to mention the software package used for the calculations, it would rather be interesting to learn more about the different physical and chemical processes modelled and considered as relevant.

The reference to Iida (2008) is missing.

References:

Aitken, J., 1890, On the number of dust particles in the atmosphere of certain places in Great Britain and on the continent with remarks on the relation between the amount of dust and meteorological phenomena, Proc. R.S.E., Vol XVII, 1889-1890.

Deng, C., Cai, R., Yan, C., Zheng, J., and Jiang, J.: Formation and growth of sub-3 nm particles in megacities: impacts of background aerosols, Faraday Discussions, 10.1039/D0FD00083C, 2020a.

Deng, C., Fu, Y., Dada, L., Yan, C., Cai, R., Yang, D., Zhou, Y., Yin, R., Lu, Y., Li, X., Qiao, X., Fan, X., Nie, W., Kontkanen, J., Kangasluoma, J., Chu, B., Ding, A., Kerminen, V., Paasonen, P., Worsnop, R. D., Bianchi, F., Liu, Y., Zheng, J., Wang, L., Kulmala, M., and Jiang, J.: Seasonal characteristics of new particle formation and growth in urban Beijing, Environmental Science and Technology, 54, 8547–8557, 10.1021/acs.est.0c00808, 2020b.

Junkermann, W. Platt, U. and Volz, A. , A Photoelectric Detector of the Measurement of Photolysis Frequencies of Ozone and other Atmospheric Molecules, Journal of Atmospheric Chemistry, 8, 203-227, 1989

Junkermann, W., Vogel, B., and Sutton, M.A., 2011, The climate penalty for clean fossil fuel combustion, Atmos. Chem. Phys, 11, 12917-12924,

Junkermann, W., and Hacker, J.M., Ultrafine particles over Eastern Australia: an airborne survey, Tellus B, 2015, 67, 25308, http://dx.doi.org/10.3402/tellusb.v67.25308

Junkermann W., and Hacker J., 2018, Ultrafine particles in the lower troposphere: major sources, invisible plumes and meteorological transport processes, BAMS, 99, 2587-2622, Dec. 2018, DOI:10.1175/BAMS-D-18-0075.1

Kerminen, V.-M., Chen, X., Vakkari, V., Petäjä, T., Kulmala, M., and Bianchi, F.: Atmospheric new particle formation and growth: review of field observations, Environmental Research Letters, 13, 10.1088/1748-9326/aadf3c, 2018.

Kulmala, M. Kerminen,V.M. Petäjä, T., Ding A.J. and Wang, L. Atmospheric gas-to-particle conversion: why NPF events are observed in megacities? Faraday Discussions, 2017, 200, 271

Mohnen V.A., and Lodge J.P. ,General survey of gas-to-particle conversions Proc. 7th ICCN, Prague (1969)

Rönkkö, T., Kuuluvainen, H., Karjalainen, P., Keskinen, J., Hillamo, R., Niemi, J. V., Pirjola, L., Timonen, H. J., Saarikoski, S., Saukko, E., Järvinen, A., Silvennoinen, H., Rostedt, A., Olin, M., Yli-Ojanperä, J., Nousiainen, P., Kousa, A., and Dal Maso, M.: Traffic is a major source of atmospheric nanocluster aerosol, P. Natl. Acad. Sci., 114, 7549–7554, https://doi.org/10.1073/pnas.1700830114, 2017.

Stevens, R. G., Pierce, J. R., Brock, C. A., Reed, M. K., Crawford, J. H., Holloway, J. S., Ryerson, T. B., Huey, L. G., and Nowak, J. B.: Nucleation and growth of sulfate aerosol in coal-fired power plant plumes: sensitivity to background aerosol and meteorology, Atmos. Chem. Phys., 12, 189–206, https://doi.org/10.5194/acp-12-189-2012, 2012.

---

## Referee Report (RR2)

**Review of Cai et al: Impacts of coagulation on the appearance time method for sub-3 nm particle growth rate evaluation and their corrections (V4)**

The paper describes a correction for the consideration of coagulation in nucleation aerosol formation modelling. This correction is changing the timing for nucleation mode particles, the time within the model until when particles have grown for example to a certain size. The title restricts this investigation to particle sizes below 3 nm where it might be especially important, however within the manuscript the appearance time is given also for other particles sizes. To illustrate the results a case study for Beijing is given for a late winter nanoparticle event.

General comments

The authors miss to clarify under which environmental conditions their theory is applicable and relevant. Page 3, line 23 they promise to investigate the limitations. However, these limitations are never specified in the text. Page 15, line 18 to 20 states: uncertainties come from instrumental biases, atmospheric turbulence and omitted contributions from transport, mixing and emission to the measured aerosol size distribution.

Summarizing the limits behind these statements mean that the theory presented is valid only for a homogeneous air-mass without any variability during the period between initialization of the model run (e.g. 00:00 until at least noon, the time given for the appearance of 4.9 nm particles (12:00), page 15, line 1). These conditions might be valid in a smog chamber (Stolzenburg et al, 2020) but are unlikely to find in the real atmosphere. Here a coupled model setup is necessary (Baklanov et al, 2008). Whether the model used in the manuscript is applicable at all for stationary ground based measurements under the conditions in the North China Plain is highly questionable.

The authors suggest by plotting model results into a graph with experimental results that the observation on February 24, 2018 can be described with their theory. However, a detailed supporting data set is missing. All available supporting data suggest that model and experiment cannot be compared as the experimental data set is not in agreement with the limitations of the model.

A three dimensional transport model including meteorological variables is available from either Flexpart of HYSPLIT. A local high resolution data set is available for example from the 320 m Beijing meteorological tower, about 7 km away from the particle observation site, close enough for a regional investigation. The data from this tower for February 24, 2018 (private communication, Prof. Fei Hu, IAP Beijing) show, that an intense vertical exchange of air masses began already a few hours before sunrise and continuing the next hours, advecting dry air from the residual layer all the way towards the ground (lowest level 8 m). This advection is reflected for example in the water vapor measurements. Starting at 2.3 g/m$^3$ the water vapor concentration declined to < 1.1 g/m$^3$ at 08:00. This is a clear indication for a massive advection of dry air from aloft (~ 0.9 - 1 g/m$^3$ in the residual layer between 800 and 1500 m above Beijing, HYSPLIT) and a replacement of the humid air in the planetary boundary layer. This air mass transport coincides with a concurrent doubling of the sulphur dioxide concentration between midnight and 08:00 LT. This polluted air mass contained also excess $H_2SO_4$, roughly doubling from nighttime levels to about 4* 10$^6$ /cm$^3$ at the time of sunrise (06:56 LT). Such an increase, even before any uv-radiation would be available for photochemistry, is not at all in agreement with a photochemical production of $H_2SO_4$ as claimed in the manuscript (Hearth et al, 2004).

The residual layer often contains large quantities of nanoparticles as shown by Quan et al, (2017) for Beijing, by Lampilahti et al, (2020) for Hyytiälä, Finland (also Hao et al, 2018) and by Junkermann and Hacker, (2018) for Germany (see there also for further examples from China and Australia).

Finally: The authors present an analysis how particles might be produced in the atmosphere in a homogeneous air parcel. An admixture of air from the outside either by a replacement of the air mass by horizontal transport or a convective mixing process with air from the residual layer (Lampilahti et al, 2020) is not taken into account. Such an approach is not in agreement with highly variable ambient conditions, typical winds and diurnal cycles in the planetary boundary layer and thus not applicable for the Beijing case study.

The case study however, properly analyzed, would be a good example for a transport and convection driven nanoparticle advection, including gas to particle conversion (Gillani et al, 1979). Such a nanoparticle advection can also superpose the physical constrains of nano-particle GTP under heavily polluted conditions (Kulmala et al, 2017).

References

Baklanov A., Mahura A., Sokhi R. (Eds) (2008): Integrated Systems of Meso-Meteorological and Chemical Transport Models. Report of the COST Action 728 on "Enhancing mesoscale meteorological modelling capabilities for air pollution and dispersion applications". COST Publication, 178p, Feb 2008

Heard, D.E. et al, Geophysical Research Letters, 2004, High levels of the hydroxyl radical in the winter urban troposphere, Volume 31, 18, https://doi.org/10.1029/2004GL020544

Gillani, N.V., Husar, R.B., Husar, J.D., Patterson D.E., and Wilson, W.E., 1978, Project MISTT, Kinetics of particulate sulphur formation in a power plant plume out to 300 km, Atmospheric Environment, 12, 589-598

Hao, L., et al, 2018, Combined effects of boundary layer dynamics and atmospheric chemistry on aerosol composition during new particle formation periods, Atmos. Chem. Phys., 18, 17705–17716, https://doi.org/10.5194/acp-18-17705-2018.

Junkermann W., and Hacker J., 2018, Ultrafine particles in the lower troposphere: major sources, invisible plumes and meteorological transport processes, BAMS, 99, 2587-2622, Dec. 2018, DOI:10.1175/BAMS-D-18-0075.1

Kulmala, M. et al, 2017, Atmospheric gas-to-particle conversion: why NPF events are observed in megacities? Faraday Discuss., 200, 271-288

Lampilahti, J., et al, Aerosol particle formation in the upper residual layer, Atmos. Chem. Phys. Discuss., https://doi.org/10.5194/acp-2020-923, in review, 2020.

Quan, J., et al, 2017: Anthropogenic pollution elevates the peak height of new particle formation from planetary boundary layer to lower free troposphere, Geophys. Res. Lett., 44(14), 7537–7543, doi:10.1002/2017GL074553

Stolzenburg, D. et al, 2020, Enhanced growth rate of atmospheric particles from sulfuric acid, Atmos. Chem. Phys., 20, 7359–7372, https://doi.org/10.5194/acp-20-7359-2020

---

## Author Response (AR3)

**Responses to Reviewer #1's Comments on Manuscript acp-2020-398**

(Impacts of coagulation on the appearance time method for new particle growth rate evaluation and their corrections)

We appreciate the efforts of Prof. Dr. Wolfgang Junkermann (referred to as the reviewer below), in elucidating the influence of meteorological factors on the measured evolution of new particles during the case study presented in Figure 8. The reviewer provided convincing analysis to show there was a non-negligible influence of transport on the local air mass. Some concerns were raised on the validity to use a box model for new particle formation in a complex urban environment. However, these concerns are based on misinterpretations of this study because the models in this study are only used to test formulae. The meteorological analysis presented by the reviewer is not against the correction of the appearance time method. In this response, we address the reviewer's comments and again clarify the objectives and main findings of this manuscript. The comments are shown as sans-serif dark red texts and our responses are shown as serif black texts. References are given at the end of the responses.

**Reviewer's comments**

The paper describes a correction for the consideration of coagulation in nucleation aerosol formation modelling. This correction is changing the timing for nucleation mode particles, the time within the model until when particles have grown for example to a certain size. The title restricts this investigation to particle sizes below 3 nm where it might be especially important, however within the manuscript the appearance time is given also for other particles sizes. To illustrate the results a case study for Beijing is given for a late winter nanoparticle event.

**Response**: This manuscript introduces a correction for a formula to calculate the new particle growth rate, not a model to simulate new particle formation. As clarified in the manuscript and previous responses, the models are used to test the corrected formula and the case study measured in urban Beijing is used as an instance for the influence of coagulation on the appearance time in polluted atmospheric environments.

Although the first part of the model description looks reasonably for an application for chamber studies the authors fail to prove that the model is applicable for the real atmosphere. The second part, the atmospheric experimental case study is not in agreement with the particle event classification. It is clearly and temporarily out of range for a 'classical' new particle formation event according to an independent meteorological analysis which is totally missing in the manuscript.

The model and the experiment are not really directly related to each other and thus should not be published in one manuscript as related to the same process.

**Response**: We agree with the reviewer that the models and the measured new particle formation event in urban Beijing are not directly related to each other and the models used in this study are not directly applicable to the real atmosphere. This is because the models are used to test the formulae rather than to simulate an atmospheric new particle formation process. We have clarified in Section 3.1 that "Note that the aerosol dynamic models are only used to provide a benchmark to compare the conventional and corrected appearance time methods in this study".

This study does not investigate new particle formation mechanisms in urban Beijing by comparing modeling results with measured data. Instead, it is aimed to improve the estimation of the growth rate of new particles close to the cluster size. The new particle growth rate is a key parameter to characterize new particle formation and it is often used to indicate the growth mechanism. However, as summarized in the Introduction, it is challenging to estimate the particle growth rate down to the cluster size, especially in the real atmosphere. Representative time methods, e.g., the appearance time method and the maximum concentration method, are favored because of their feasibility to report a value (e.g., Kulmala et al., 2013; Bianchi et al., 2016; Yao et al., 2018), yet the demonstration for the relationship between this reported value and the particle growth rate needs demonstration in addition to intuition. The derivation of the appearance time method in an ideal condition with constant vapor production rate and negligible sinks was reported in a quite recent study (He et al., 2020). In this study, we show that the inevitable violation of this ideal condition may affect the appearance time and hence the estimated growth rate method. The influences of coagulation sink and their corrections are derived, tested by models, and reported in a case study in urban Beijing. The uncertainties caused by other factors, e.g., size-dependent vapor evaporation and the variation of vapor concentration are investigated are reported.

In the revised manuscript, we have emphasized in the Abstract and Conclusions that even after correcting the influence of coagulation, there may be uncertainties in the growth rate retrieved by the appearance time method due to the growth rate method. Note that these uncertainties do not indicate that the corrected appearance time method is completely not valid for atmospheric studies. Instead, when investigating particle growth mechanisms by comparing measured and simulated particle growth rates, accounting for these uncertainties helps to increase the robustness of conclusions.

The reviewer proves that the local air mass on Feb. 24, 2018 was significantly affected by transport based on a meteorological analysis. We will show below that this meteorological analysis is not against the findings of this study.

**General comments**

The authors miss to clarify under which environmental conditions their theory is applicable and relevant. Page 3, line 23 they promise to investigate the limitations. However, these limitations are never specified in the text. Page 15, line 18 to 20 states: uncertainties come from instrumental biases, atmospheric turbulence and omitted contributions from transport, mixing and emission to the measured aerosol size distribution.

**Response**: Instead of presenting environment conditions, we have stated the assumptions for the appearance time method at the beginning of Section 4.1. Violating these assumptions may cause uncertainties in the appearance time and hence limit the accuracy of the retrieved particle growth rate. In Sections 4.2 and 4.3, we discuss these limitations due to the violation of these assumptions, derive a correction formula to minimize the limitation due to the influence of coagulation, and report other potential limitations. We prefer to keep this assumption-violation-uncertainty discussion rather than specify several certain environmental conditions because it is essentially the population balance equations (e.g., Eq. 14) that determines the validity and potential uncertainty of the appearance time method. For instance, as stated in lines 18-20 on page 15, a non-negligible contribution of transport to the measured aerosol number concentration introduces uncertainties to the appearance time method because the third ideal assumption, i.e., "condensation is the only cause of the change in particle concentrations" is violated.

Summarizing the limits behind these statements mean that the theory presented is valid only for a homogeneous air-mass without any variability during the period between initialization of the model run (e.g. 00:00 until at least noon, the time given for the appearance of 4.9 nm particles (12:00), page 15, line 1). These conditions might be valid in a smog chamber (Stolzenburg et al, 2020) but are unlikely to find in the real atmosphere. Here a coupled model setup is necessary (Baklanov et al, 2008). Whether the model used in the manuscript is applicable at all for stationary ground based measurements under the conditions in the North China Plain is highly questionable.

The authors suggest by plotting model results into a graph with experimental results that the observation on February 24, 2018 can be described with their theory. However, a detailed supporting data set is missing. All available supporting data suggest that model and experiment cannot be compared as the experimental data set is not in agreement with the limitations of the model.

**Response**: The model used in this manuscript (e.g., in Fig. 7) is used to test the formulae for the appearance time method and it is not compared to measurement results. We have clarified this in Section 3.1.

As clarified in the caption in Fig. 8, there is no modeling result compared to the measurement results on Feb. 24, 2018. The aerosol size distribution and $H_2SO_4$ concentration in Fig. 8a were measured instruments, the measured growth rates in Fig. 8b were estimated from the measured evolution of aerosol size distributions using the conventional (Eq. 5) and corrected (Eq. 8) appearance time methods, and the theoretical growth rates were calculated using the measured $H_2SO_4$ concentration (Eq. 1) and aerosol size distributions (Eq. 2). The objective of this figure is to show that the coagulation correction for the appearance time method is necessary for sub-3 nm particles for this case study.

A three dimensional transport model including meteorological variables is available from either Flexpart of HYSPLIT. A local high resolution data set is available for example from the 320 m Beijing meteorological tower, about 7 km away from the particle observation site, close enough for a regional investigation. The data from this tower for February 24, 2018 (private communication, Prof. Fei Hu, IAP Beijing) show, that an intense vertical exchange of air masses began already a few hours before sunrise and continuing the next hours, advecting dry air from the residual layer all the way towards the ground (lowest level 8 m). This advection is reflected for example in the water vapor measurements. Starting at 2.3 g/m$^3$ the water vapor concentration declined to < 1.1 g/m$^3$ at 08:00.

This is a clear indication for a massive advection of dry air from aloft (~ 0.9 - 1 g/m3 in the residual layer between 800 and 1500 m above Beijing, HYSPLIT) and a replacement of the humid air in the planetary boundary layer. This air mass transport coincides with a concurrent doubling of the sulphur dioxide concentration between midnight and 08:00 LT. This polluted air mass contained also excess $H_2SO_4$, roughly doubling from nighttime levels to about 4* 10$^6$ /cm$^3$ at the time of sunrise (06:56 LT).

**Response**: We agree with the reviewer that a significant advection of air mass occurred before sunrise on Feb. 24, 2018. This advection is indicated by the changes in the concentrations of vapors with relatively long atmospheric residence time, e.g., $H_2O$ and $SO_2$. We hold our opinions on whether the increase of $H_2SO_4$ concentration was mainly caused by transport of $H_2SO_4$ molecules or the changes of their sources and sinks due to transport or other reasons. This is because the atmospheric residence time of $H_2SO_4$ was short (~40 s) due to the relatively high condensation sink.

Such an increase, even before any uv-radiation would be available for photochemistry, is not at all in agreement with a photochemical production of $H_2SO_4$ as claimed in the manuscript (Hearth et al, 2004).

**Response**: We did not discuss the formation mechanism of $H_2SO_4$ in this study and did not state that all the $H_2SO_4$ molecules in urban Beijing, especially during nighttime, was formed by photochemical reactions.

The residual layer often contains large quantities of nanoparticles as shown by Quan et al, (2017) for Beijing, by Lampilahti et al, (2020) for Hyytiälä, Finland (also Hao et al, 2018) and by Junkermann and Hacker, (2018) for Germany (see there also for further examples from China and Australia).

**Response**: Advection indicates a potential influence of transport on the measured aerosol size distributions. This influence is affected by the size-dependent atmospheric residence time of aerosols. Aerosols close to the cluster size are thought to be formed locally because of their corresponding high coagulation sink. In Fig. 8, the particle growth rate in the sub-10 nm size range was discussed and the influence of coagulation was found to be significant for sub-3 nm particles.

The concentration of larger nanoparticles (e.g., >25 nm) with longer atmospheric residence time may be more significantly affected by transport compared to that of sub-10 nm particles. This may impact the size-dependent population balance of aerosols and hence cause uncertainties in the retrieved particle growth rate. Several studies (e.g., Cai et al., 2018; Kontkanen et al., 2020) have tried to estimate this influence from the population balance point of view. We agree with the reviewer that the influence of transport on the measured number concentrations of large nanoparticle are sometimes important in the atmosphere and have clarified that "transport, mixing, and emissions might influence the observed aerosol size distributions" when discussing the measured aerosol size distributions on Feb. 24, 2018 (line 12, page 16).

Finally: The authors present an analysis how particles might be produced in the atmosphere in a homogeneous air parcel. An admixture of air from the outside either by a replacement of the air mass by horizontal transport or a convective mixing process with air from the residual layer (Lampilahti et al, 2020) is not taken into account. Such an approach is not in agreement with highly variable ambient conditions, typical winds and diurnal cycles in the planetary boundary layer and thus not applicable for the Beijing case study.

**Response**: We did not discuss how particles were produced in this study.

The case study however, properly analyzed, would be a good example for a transport and convection driven nanoparticle advection, including gas to particle conversion (Gillani et al, 1979). Such a nanoparticle advection can also superpose the physical constrains of nano-particle GTP under heavily polluted conditions (Kulmala et al, 2017).

**Response**: We agree with the reviewer that transport may be an important source of nanoparticles with sufficiently long atmospheric residence time in the urban atmospheric environment. The different sources (e.g., vapor concentrations) and sinks (e.g., coagulation sink) at different heights may cause non-uniform vertical distribution of nanoparticles, yet more studies are needed to verify and quantify the contributions of vertical transport to the measured aerosol size distributions at the ground level. We look forward to possible collaborative researches with the reviewer to address this scientific question.

**References**

Bianchi, F., Tröstl, J., Junninen, H., Frege, C., Henne, S., Hoyle, C. R., Molteni, U., Herrmann, E., Adamov, A., Bukowiecki, N., Chen, X., Duplissy, J., Gysel, M., Hutterli, M., Kangasluoma, J., Kontkanen, J., Kürten, A., Manninen, H. E., Münch, S., Peräkylä, O., Petäjä, T., Rondo, L., Williamson, C., Weingartner, E., Curtius, J., Worsnop, D. R., Kulmala, M., Dommen, J., and Baltensperger, U.: New particle formation in the free troposphere: A question of chemistry and timing, Science, 352, 1109-1112, 10.1126/science.aad5456, 2016.

Cai, R., Chandra, I., Yang, D., Yao, L., Fu, Y., Li, X., Lu, Y., Luo, L., Hao, J., Ma, Y., Wang, L., Zheng, J., Seto, T., and Jiang, J.: Estimating the influence of transport on aerosol size distributions during new particle formation events, Atmospheric Chemistry and Physics, 18, 16587-16599, 10.5194/acp-18-16587-2018, 2018.

He, X.-C., Iyer, S., Sipilä, M., Ylisirniö, A., Peltola, M., Kontkanen, J., Baalbaki, R., Simon, M., Kürten, A., Tham, Y. J., Pesonen, J., Ahonen, L. R., Amanatidis, S., Amorim, A., Baccarini, A., Beck, L., Bianchi, F., Brilke, S., Chen, D., Chiu, R., Curtius, J., Dada, L., Dias, A., Dommen, J., Donahue, N. M., Duplissy, J., El Haddad, I., Finkenzeller, H., Fischer, L., Heinritzi, M., Hofbauer, V., Kangasluoma, J., Kim, C., Koenig, T. K., Kubečka, J., Kvashnin, A., Lamkaddam, H., Lee, C. P., Leiminger, M., Li, Z., Makhmutov, V., Xiao, M., Marten, R., Nie, W., Onnela, A., Partoll, E., Petäjä, T., Salo, V.-T., Schuchmann, S., Steiner, G., Stolzenburg, D., Stozhkov, Y., Tauber, C., Tomé, A., Väisänen, O., Vazquez-Pufleau, M., Volkamer, R., Wagner, A. C., Wang, M., Wang, Y., Wimmer, D., Winkler, P. M., Worsnop, D. R., Wu, Y., Yan, C., Ye, Q., Lehtinen, K., Nieminen, T., Manninen, H. E., Rissanen, M., Schobesberger, S., Lehtipalo, K., Baltensperger, U., Hansel, A., Kerminen, V.-M., Flagan, R. C., Kirkby, J., Kurtén, T., and Kulmala, M.: Determination of the Collision Rate Coefficient between Charged Iodic Acid Clusters and Iodic Acid using the Appearance Time Method, Aerosol Science and Technology, 1-17, 10.1080/02786826.2020.1839013, 2020.

Kulmala, M., Kontkanen, J., Junninen, H., Lehtipalo, K., Manninen, H. E., Nieminen, T., Petäjä, T., Sipilä, M., Schobesberger, S., Rantala, P., Franchin, A., Jokinen, T., Jarvinen, E., Äijälä, M., Kangasluoma, J., Hakala, J., Aalto, P. P., Paasonen, P., Mikkilä, J., Vanhanen, J., Aalto, J., Hakola, H., Makkonen, U., Ruuskanen, T., Mauldin, R. L., 3rd, Duplissy, J., Vehkamäki, H., Bäck, J., Kortelainen, A., Riipinen, I., Kurtén, T., Johnston, M. V., Smith, J. N., Ehn, M., Mentel, T. F., Lehtinen, K. E., Laaksonen, A., Kerminen, V.-M., and Worsnop, D. R.: Direct observations of atmospheric aerosol nucleation, Science, 339, 943-946, 10.1126/science.1227385, 2013.

Yao, L., Garmash, O., Bianchi, F., Zheng, J., Yan, C., Kontkanen, J., Junninen, H., Mazon, S. B., Ehn, M., Paasonen, P., Sipilä, M., Wang, M., Wang, X., Xiao, S., Chen, H., Lu, Y., Zhang, B., Wang, D., Fu, Q., Geng, F., Li, L., Wang, H., Qiao, L., Yang, X., Chen, J., Kerminen, V.-M., Petäjä, T., Worsnop, D. R., Kulmala, M., and Wang, L.: Atmospheric new particle formation from sulfuric acid and amines in a Chinese megacity, Science, 361, 278-281, 10.1126/science.aao4839, 2018.